



# Research on 16-day Planetary Waves in the Mid-latitude Troposphere, Stratosphere, Mesosphere, and Lower Thermosphere with Langfang Dual-frequency ST-M Radar Data

Zengmao Zhang[1,2,3], Xiong Hu[1,3], Qingchen Xu[1,3], Bing Cai[1,3], Junfeng Yang[1,3]

[1] National Space Science Center, Chinese Academy of Sciences, Beijing 100190, China
[2] University of Chinese Academy of Sciences, Beijing 100049, China
[3] State Key Laboratory of Space Weather, Beijing 100190, China

*Correspondence to*: Xiong Hu (xhu@nssc.ac.cn)

**Abstract.** Horizontal wind observational data by the dual-frequency Stratosphere-Troposphere-Meteor (ST-M) radar at
Langfang Observatory from March 2023 to February 2024, was used to investigate spatiotemporal variations, and
propagation characteristics of planetary waves, as well as the relationship between planetary waves in the troposphere and
stratosphere (ST) and the mesosphere and lower thermosphere (MLT) over Langfang mid-latitude regions. The quasi-16-day
planetary wave's activities are obtained by applying band-pass filtering on the daily averaged horizontal wind. Simultaneous
MERRA-2 reanalysis wind data are used to derive the dominant zonal wavenumbers of 16-day waves in ST and mesosphere,
and also the background zonal winds through which the planetary may propagate vertically.  Results show that 16-day wave
activity occurs all the year, its zonal component is stronger than the meridional component, and it is characterized by being
strong in winter and weak in summer.  It is newly found that the vertical phase propagation direction of 16-day wave got
changed during autumn and winter that in autumn August-September it is upward in ST and downward in MLT, and upward
in ST but upward in MLT in November-December, and downward in ST and upward in MLT after later December. The
dominant zonal wavenumbers for the 16-day wave are (ST: -1, MLT: 2) in August-September, and (ST: 2, MLT: 4) in
November-December, and (ST: -1, MLT: 4) in December-January respectively in MERRA-2 data.  It can be derived with the
information of vertical phase velocity and zonal wavenumber that the group velocity of the 16-days in radar data is
downward in ST and downward in MLT in August-September, and upward in ST and upward in MLT in November-
December, and downward in ST and Upward in MLT in December-January, respectively. Together with the zonal
background winds from MERRA-2 and radar over the field site which provide the vertical propagation condition for
planetary waves, it can infer that the observed 16-day wave in ST may be triggered by the jet at about 14km altitude and
hence propagated downward in August-September, and the background wind do not allow upward propagating of the wave.
So, the observed wave in MLT in August-September may be trigged by another unknown source above or refracted from
low-or-high latitude regions. The observed 16-day wave in ST in November-December is not the same as that before, was
generated in the lower atmosphere and propagated through the background winds upward maybe into the MLT regions as
observed. In December-January, the observed 16-day wave in ST gets changed zonal wavenumber again, it is also generated





in the lower atmosphere and propagate upward. However, its upward propagation will be blocked by the above winds and therefore cannot penetrate into the MLT above. The observed wave in MLT in December-January could be the one already existed there before. The newly observations and interpretations help us to further understanding the vertical coupling

among the ST and MLT by planetary waves.

**Keywords**. ST-M radar, Troposphere and Stratosphere, Mesosphere and Lower Thermosphere, 16-day planetary waves, vertical propagation

## 1 Introduction

Planetary waves are ubiquitous across various latitudes on Earth, with periods typically ranging from several days to tens of days and horizontal wavelengths extending from thousands to tens of thousands of kilometres. They are a key component of atmospheric oscillations, significantly influencing atmospheric circulation patterns and the distribution of matter. Variations in their amplitude may be linked to certain extreme weather events, highlighting their importance in atmospheric research (de Vries et al., 2024; Frank, Galinsky, Zhang, & Ralph, 2024; Lekshmi, Chattopadhyay, & Pai, 2024). According to the

propagation characteristics of planetary waves, they can be divided into static planetary waves and traveling planetary waves. Static planetary waves are mainly influenced by topographical differences, such as land-sea distribution, and their phase is relatively fixed (Charney & Eliassen, 1949; Smagorinsky, 1953); traveling planetary waves exhibit propagation characteristics, typically propagating periodically eastward or westward along the latitude circles, with their phase changing at different times (Fedulina, Pogoreltsev, & Vaughan, 2004; Zheng, Yun, & Shaodong, 2024). Planetary waves with

oscillation periods of 1.7–3.4 days, 4.5–6.2 days, 7.5–12 days, and 12–20 days are referred to as quasi-2-day, quasi-5-day, quasi-10-day, and quasi-16-day waves, respectively (Merzlyakov, Solovjova, Yudakov, & Physics, 2013; Portnyagin et al., 1999; Murry L. Salby, 1981). Planetary waves are primarily generated in the troposphere (Dickinson, 1968; J. M. Forbes et al., 1995). However, studies have shown that significant planetary wave activity can also be observed in the stratosphere, as well as in the mesosphere and lower thermosphere (MLT) (Charney & Drazin, 1961; Eliassen, 1960; Ronghui HUANG,

55    2018).

In recent years, research on the quasi-16-day planetary waves using atmospheric radar observation data has received widespread attention(Day & Mitchell, 2010; J. M. Forbes et al., 1995; Gong et al., 2020; Guharay, Batista, Clemesha, Buriti, & Schuch, 2016; Luo et al., 2002; Yang et al., 2023) . The findings indicate that the zonal component of 16-day waves is generally significantly stronger than the meridional component at different latitudes and altitudes. However, there are

notable differences in their seasonal characteristics, zonal wavenumbers and propagation directions, source locations, and vertical propagation directions.

The seasonal characteristics of the 16-day wave are related to latitude and altitude. In the MLT region of mid- and high-latitudes, the 16-day wave typically exhibits significant seasonal variations, but the patterns of these changes are not





consistent. Most studies indicate that the 16-day wave in the MLT is strongest in winter and weaker in summer(Day & Mitchell, 2010; J. M. Forbes et al., 1995; Gong et al., 2020; Yang et al., 2023). For example, Gong et al. observed that the 16-day wave in the MLT over three stations—Mohe (53.5° N, 122.3° E), Beijing (40.3° N, 116.2° E), and Wuhan (30.5° N, 114.6° E)—exhibited the characteristic of being strong in winter and weak in summer(Gong et al., 2020); Yang et al.'s observational study in Mengcheng (33.4° N, 116.5° E) and Day et al.'s observational studies at Esrange (68° N, 21° E) and Rothera (68° S, 68° W) both revealed the same seasonal characteristics(Day & Mitchell, 2010; Yang et al., 2023). There are also studies that differ from the above conclusions. For example, Williams et al. observed a strong 16-day wave during summer in Alaska (65° N, 147° W) (Williams & Avery, 1992); Jiang et al.'s study in Wuhan (30.5° N, 114.3° E) revealed that the 16-day wave is strongest in autumn, with no significant 16-day wave activity observed in winter(Guoying Jiang, 2005). In the MLT region at low latitudes, the 16-day wave does not exhibit significant seasonal variations(Araújo, Lima, Batista, Clemesha, & Takahashi, 2014; Gaikwad et al., 2023; Lima, Batista, Clemesha, & Takahashi, 2006; Namboothiri, Kishore, & Igarashi, 2002). In the ST region, the 16-day wave is observed throughout the year in the troposphere, while in the stratosphere, it is mainly concentrated in the winter(C. Huang, Zhang, Chen, Zhang, & Huang, 2017; Lei Tang 2016; Williams & Avery, 1992).

The zonal wavenumber and propagation direction of the 16-day wave are influenced by latitude, height, and zonal background wind. Its zonal phase velocity propagates westward relative to the zonal background wind, while the zonal group velocity relative to the zonal background wind can propagate both westward and eastward(O'Neill, 1989). In classical planetary wave theory, the 16-day wave corresponds to the second symmetric Rossby mode with a wavenumber $k = 1$, propagating westward(J. M. Forbes et al., 1995; M. L. Salby, 1981). However, with the further study of the 16-day wave, more westward and eastward propagating 16-day wave modes have been observed(X. S. Huang et al., 2022; Kleinknecht, Espy, & Hibbins, 2014; Shepherd & Tsuda, 2008; Yu et al., 2019). For example, Tang et al. used MST radar data from Hebei Xianghe (39.8° N, 116.9° E) along with MERRA reanalysis data. They pointed out that, based on phase relationships at different sites, the 16-day wave had a zonal wavenumber of 2 and propagated eastward from west to east in the 6-24 km height range during February-March 2014(Lei Tang 2016); Huang et al. also used MST radar observation data from Hebei Xianghe and MERRA data to derive the frequency-wavenumber spectrum. They found that from December 2013 to November 2014, the predominant zonal wavenumbers of the quasi-16-day wave were 2 and 3, with almost all of them propagating from west to east(C. Huang et al., 2017); Huang, using meteor radar data from Beijing (40.3° N, 116.2° E) and Wuhan (30.5° N, 114.6° E) along with MERRA data, derived the frequency-wavenumber spectrum. They found that from late November to December 2013, the zonal wavenumber of the 16-day wave in the stratosphere to MLT (Mesosphere-Lower Thermosphere) region was 1, and from late January to late February 2014, it was 2. In both cases, the waves propagated from east to west(X. S. Huang et al., 2022).

The wave sources of the 16-day wave vary at different altitudes and require further determination of their locations by considering the background wind and vertical propagation conditions. The troposphere contains abundant excitation sources, such as terrain inhomogeneities and solar thermal radiation, and is considered the source region for planetary waves;





Stratospheric planetary waves often appear in winter, and their main sources are likely twofold: 1. Vertical upward propagation of planetary waves generated in the troposphere; 2. Meridional propagation of stratospheric planetary waves

from high-latitude regions(Jeffrey M. Forbes, 1995; Lei Tang 2016; Williams & Avery, 1992). For example, Tang et al. used Lomb-Scargle (LS) spectral analysis to study the 16-day wave above Hebei Xianghe (39.8° N, 116.9° E). They observed that sometimes the appearance of stratospheric planetary waves coincided with those in the troposphere, while at other times, strong wave activity was only observed in the stratosphere. They suggested that this could be due to the simultaneous presence of two distinct sources of planetary waves. By combining MERRA reanalysis data, they pointed out that in the 6-24

km height range during February-March 2014, the phase surfaces of the 16-day wave had a positive slope. This observation suggests that the wave's energy was propagating vertically upward(Lei Tang 2016). Huang et al., based on the phase of the 16-day wave amplitude in the Hebei Xianghe ST region, observed a positive slope in December 2013, January 2014, and May 2014. This led them to conclude that the energy was propagating vertically upward in these periods. They also discussed the vertical propagation conditions through the quasi-geostrophic refractive index squared. Their analysis showed

that regions in the lower troposphere (3.5-5 km) and near the tropopause (15-20 km) would inhibit the vertical propagation of the 16-day wave. This finding is consistent with the observed variation of the 16-day wave amplitude with height(C. Huang et al., 2017). The 16-day wave in the MLT region, before and after winter, is most likely sourced from three main aspects: 1. The wave source is typically located in the lower atmosphere, propagating vertically upward to the MLT; 2. The wave is generated locally within the MLT region; 3. It may originate from the propagation of waves from the winter

hemisphere(Charney & Drazin, 1961; Espy, Stegman, & Witt, 1997; J. M. Forbes et al., 1995; Luo et al., 2002; Miyoshi, 1999; M. L. Salby, 1981; Murry L. Salby, 1981; Venkat Ratnam, 2024; Williams & Avery, 1992; Yu et al., 2019). For example, Forbes et al., using medium-frequency radar observation data from Obninsk (54° N, 38° E) and Saskatoon (52° N, 107° W), found a wavenumber 1 quasi-16-day wave at 95 km in January-February 1979. Numerical simulations showed that this 16-day wave was consistent with the upward propagation of the 16-day wave in the troposphere and stratosphere. This

confirmed that the 16-day wave in the winter MLT could propagate directly upward from the lower atmosphere. Numerical simulation results of the background wind conditions also showed that the contour lines of wind amplitude at the equatorial mesopause were not completely closed, indicating the presence of a meridional channel between the hemispheres. The authors suggested that the 16-day wave observed in the MLT during summer could be the result of the 16-day wave from the winter hemisphere leaking through this meridional channel. At the same time, they also proposed the possibility that the 16-

day wave in the stratosphere could modulate gravity waves, which would deposit momentum in the MLT region, thereby generating the 16-day wave in the MLT(J. M. Forbes et al., 1995). Gong et al. calculated the quasi-geostrophic refractive index squared at different altitudes for the years 2008 to 2017 at sites including Mohe (53.5° N, 122.3° E), Beijing (40.3° N, 116.2° E), and Wuhan (30.5° N, 114.6° E). Their results indicated: Over Beijing, the quasi-geostrophic refractive index squared was positive around March and October, meeting the vertical propagation conditions, allowing the 16-day wave in

the troposphere and stratosphere to propagate upward to the MLT; In Wuhan, around 30 km, vertical propagation conditions were generally not satisfied throughout the year. The 16-day wave in the MLT primarily originated from the mesosphere



around March and October; In Mohe, during winter, the refractive index squared was mostly positive below 60 km, suggesting that the 16-day wave could propagate from the troposphere to the mesosphere in winter. However, in the 55-60 km region, vertical propagation conditions were not met for most of the time, implying that the 16-day wave in the MLT

during this period did not originate from upward propagation of the wave in the lower atmosphere. Gong et al. proposed that inter-hemispheric propagation could be the mechanism for the generation of the 16-day wave in the Mohe MLT region(Gong et al., 2020). When determining the vertical propagation direction of planetary waves, some studies assume that the phase propagation direction of Rossby waves is opposite to the energy propagation direction. However, this assumption does not fully account for the influence of the zonal propagation direction on the vertical direction of planetary waves(Guoying Jiang,

2005; C. Huang et al., 2017; Lei Tang 2016).

The studies of the 16-day wave using radio atmosphere radar mentioned above primarily focus on specific regions within the ST or MLT. Yu et al., using observation data from three meteor radars in Wuhan (30.5° N, 114.6° E), Beijing (40.3° N, 116.2° E), Mohe (52.5° N, 122.3° E), and an MST radar in Chongyang (29.3° N, 114.8° E), analysed the quasi-16-day wave activity in the stratosphere and MLT during the final warming period. They found that the 16-day wave was strong from the

troposphere to the MLT across different latitudes during the final warming period, with its intensity decreasing as the latitude increased. The Chongyang MST radar is located about 100 km from the Wuhan meteor radar, but the study did not analyse the relationship between the 16-day waves observed in the ST and MLT by these two radars(Yu et al., 2019).

So far, there is a relatively comprehensive understanding of the 16-day wave at different latitudes and altitudes. However, the seasonal characteristics of the 16-day wave in mid-latitude regions, as well as the relationship between vertical

propagation direction and zonal wavenumber, still require further research. In addition, previous studies have not yet utilized radio atmosphere radars at the same site to simultaneously observe and conduct in-depth analysis of the 16-day wave in both the ST and MLT regions. The newly constructed dual-frequency ST-M radar at the Langfang Observation Station (39.39° N, 116.66° E), part of the National Space Science Centre, Chinese Academy of Sciences, provides simultaneous horizontal wind field data for both the ST and MLT regions. This radar can be used to study the 16-day wave activity characteristics

above Langfang in both regions. This study conducts a statistical analysis of the daily average horizontal wind field structure and the variation of planetary waves with time and height above Langfang from March 2023 to February 2024. The analysis focuses on two altitude ranges: near the ground to 18 km and 80 to 100 km. The research emphasizes the seasonal characteristics, source locations, propagation characteristics of the 16-day planetary wave, and the relationship between planetary waves in the two regions.

**2 Data Sources and Analysis Methods**

To investigate the spatiotemporal variations, propagation characteristics, and the relationship between 16-day planetary waves in the MLT and ST, horizontal wind data from the dual-frequency ST-M radar and the NASA global atmospheric reanalysis dataset MERRA-2 were analysed ("EARTHDATA, NASA Goddard Earth Sciences (GES) Data and



Information Services Center (DISC). https://disc.gsfc.nasa.gov/,"). MERRA-2 integrates satellite observation data, ground
meteorological observations, and climate model outputs for reanalysis, combining multiple data sources. Its model-level
reanalysis dataset includes physical parameters such as zonal and meridional wind speed, pressure, and temperature across
72 model levels, defined using pressure coordinates and extending from the surface to approximately 80 km in altitude. The
temporal resolution of the data is 3 hours (Gelaro et al., 2017; C. M. Huang et al., 2021).

The dual-frequency ST-M radar at the Langfang Observatory can alternately operate in the 53.8 MHz stratosphere-
troposphere mode and the 35.0 MHz meteor mode (Xu et al., 2024). Table 1 provides an overview of its basic parameters.

**Table 1 Basic parameters of the dual-frequency ST-M radar at Langfang Observatory**

| Parameter | Value |
|---|---|
| Location | Langfang Observatory, National Space Science Center, Chinese Academy of Sciences (39.39° N, 116.66° E) |
| Operating frequency | Meteor mode 35.0Mhz, ST mode 53.8MHz |
| Peak power | 48kW |
| Number of channels | 6 (5 meteor reception channels, 1 ST reception channel) |
| Observation mode | ST low mode/high mode, meteor mode, interlaced mode |
| Detection height | Meteor mode 70–110km<br>ST low mode 1.2–8km, ST high mode 1.2–22.2km |

The horizontal wind data (after removing outliers) from the dual-frequency ST-M radar at the Langfang Observatory,
covering the period from March 1, 2023, to February 29, 2024, were selected to analyse planetary wave propagation
characteristics. These data were combined with MERRA-2 data for the 39.5° N latitude band. The meteor-mode data from
the ST-M radar have a spatial resolution of 2 km and a temporal resolution of 1 hour, while the ST-mode data have a spatial
resolution of 0.6 km and the same temporal resolution. The ST-M radar obtains wind data using both turbulent echo signals
and meteor trail echo signals. As altitude increases, the intensity of atmospheric turbulence echoes decreases, and the
meteors detected by the radar are distributed in the 80–100 km range. Considering data continuity and coverage, this study
focuses on the altitude ranges of 1.2–18 km and 80–100 km.

To obtain the temporal distribution of the 16-day planetary wave intensity in the MLT and ST, as well as the phase of the 16-
day planetary wave and the zonal background wind in the MLT, the following methods were used to analyse the planetary
wave characteristics from the ST-M radar data:

(1) Extracting Daily Mean Wind: The 24-hour data for each day is averaged to obtain the daily mean horizontal wind. This
process helps eliminate interference from tides and other short-period fluctuations when extracting planetary waves (Y. Luo
et al., 2002). If the wind data for a given day exceeds 10 hours and is evenly distributed throughout the day, the daily mean
horizontal wind data for that day is considered valid. After removing invalid data, the temporal coverage of the daily mean
horizontal wind data is shown in Table 2.





**Table 2 Time coverage of wind data by month from Mar. 2023 to Feb. 2024(Percentage)**

| Month / Altitude | Mar. | Apr. | May | June | July | Aug. | Sept. | Oct. | Nov. | Dec. | Jan. | Feb. |
|---|---|---|---|---|---|---|---|---|---|---|---|---|
| 80~100km | 77 | 87 | 100 | 100 | 87 | 52 | 100 | 100 | 100 | 100 | 100 | 100 |
| 1.2~18km | 77 | 87 | 100 | 100 | 87 | 52 | 100 | 100 | 100 | 100 | 100 | 100 |

(2) Detrending: A sliding window is applied to segment the daily average horizontal wind data in both regions. For planetary waves, the window size is set to 32 days, with a step size of 2 days, allowing for the observation of temporal variations in planetary waves. If the amount of data within the sliding window is less than 50%, the segment is considered invalid. To account for potential larger periodic seasonal variations in the atmosphere, the data within the sliding window are first detrended using a least square fitting formula (y = at + b) to obtain the wind perturbations.

(3) Spectral Analysis: A sliding window of 32 days with a step size of 10 days was used to apply Lomb-Scargle spectral analysis to the wind perturbations within the window. If the amount of data within the sliding window is less than 50%, the segment is considered invalid. This analysis provides the distribution of planetary wave intensity with different periods over time. The Lomb-Scraggle method is a statistical technique for detecting periodic signals and extracting features, particularly suited for handling unevenly sampled data. It allows for direct periodic analysis on the raw data without the need for interpolation.

(4) Band-pass Filtering: A band-pass filter with a frequency range of 12–20 days was applied to the daily average horizontal wind data to extract the spatiotemporal variation characteristics of the quasi-16-day wave. A Butterworth filter, known for its maximally flat frequency response within the passband and absence of ripples, was used. Before filtering, gaps in the data were filled using the nearest neighbour interpolation method, and after filtering, results for continuous periods with missing data were removed. This approach enhances the stability of the filtering process while ensuring the physical validity of the final analysis.

(5) 16-day Wave Harmonic Fitting: A 32-day sliding window was used to perform harmonic fitting on the filtered results of the 16-day wave, providing the phase of the 16-day planetary wave and the zonal background wind (with the 16-day wave perturbations removed) in the 80–100 km range. The sliding window step size was set to 2 days. If more than 50% of the data within the window was missing, the segment was considered invalid. The general form of a single harmonic function can be expressed as:

$$f(t) = C + A\cos(2\pi f t - \phi)\,,\tag{1}$$

Where $C$ is the offset, $A$ is the amplitude, $f$ is the frequency (which is related to the period $T$ by $f = 1/T$), and $\phi$ is the phase.

To extract the zonal wavenumber of the 16-day planetary waves and the zonal background wind from the surface to approximately 80 km, the following method was used to analyse planetary wave characteristics from the MERRA-2 data:





(1) Extracting Daily Mean Wind: The temporal resolution of the MERRA-2 data is 3 hours. The daily mean wind is calculated by averaging the 8 data points at each altitude over the course of the day.

(2) Detrending: The daily mean wind extracted from the MERRA-2 reanalysis data over each studied period (32 days per period in the study) was detrended to obtain the zonal wind perturbations.

(3) 2D Fourier Transform: A two-dimensional Fourier transform is applied to the zonal wind perturbations along the latitude band to obtain the frequency-wavenumber spectrum, from which the zonal wavenumber of the 16-day planetary wave is extracted.

(4) Calculation of Vertical Propagation Conditions: The condition for vertical propagation of planetary waves is satisfied when $0 < \bar{u} - c_x < U_c$, where $\bar{u}$ is the zonal mean wind (positive eastward), $c_x$ is the phase speed of the wave, and $U_c$ is the

critical speed for planetary waves. Using temperature and pressure data from the MERRA-2 reanalysis, the critical speed for the 16-day wave is calculated using equation (2):

$$U_c = \beta / [(k^2 + l^2) + f_0^2 / 4H^2 N^2] \ , \tag{2}$$

Where $f_0$ is the Coriolis parameter, $\beta$ represents the rate of change of $f_0$ with latitude, $k$ and $l$ are the zonal and meridional wavenumbers, respectively, $H$ is the atmospheric scale height, and $H$ is the Brunt-Väisälä frequency.

(5) 16-day Wave Harmonic Fitting: Harmonic fitting was applied to the MERRA-2 reanalysis data using a 32-day window with a 2-day step size to obtain the zonal background wind from the surface to approximately 79 km, after removing the 16-day wave fluctuations. Based on the conditions for the vertical propagation of planetary waves, the relationship between planetary waves in the MLT and ST was discussed.

**3 Results and Discussion**

**3.1 Results of the daily average horizontal wind**

Figure 1 shows the daily average horizontal wind at 80–100 km over Langfang from March 2023 to February 2024. The horizontal axis represents time (divided by months), the vertical axis represents altitude (km), and wind speed is measured in m/s. Positive values for the meridional wind indicate southerly wind, and positive values for the zonal wind indicate westerly wind. Blank areas represent missing data.



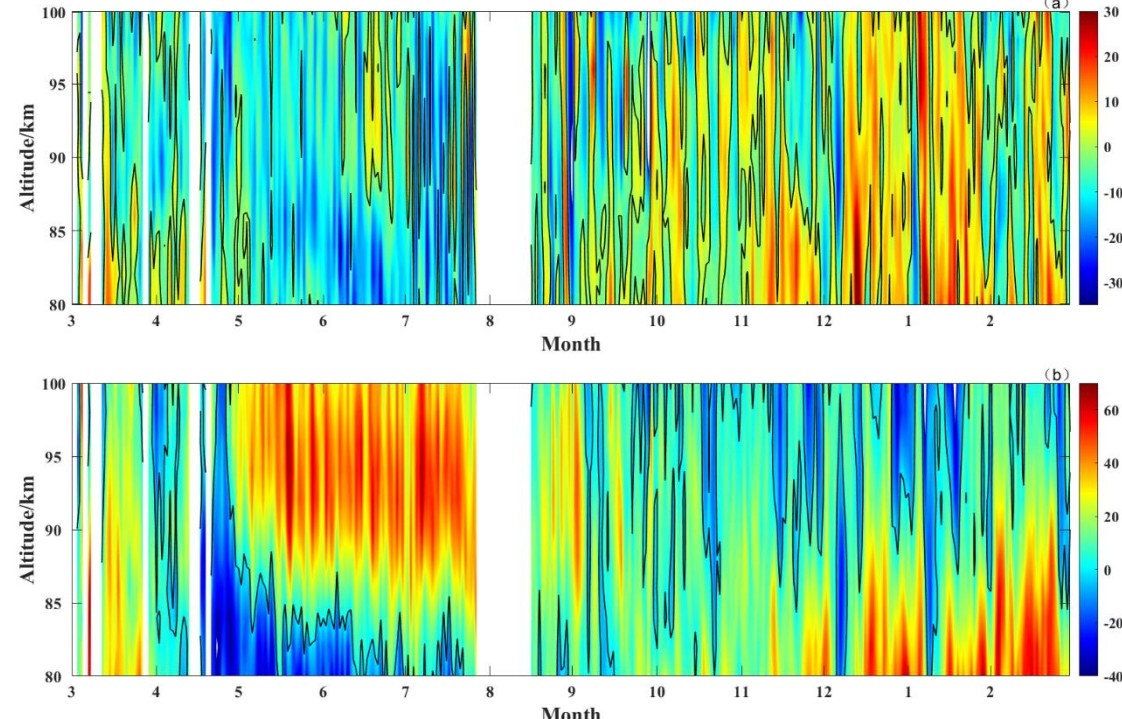

**Figure 1: Daily average horizontal wind at 80–100 km over Langfang from March 2023 to February 2024: (a) Daily average meridional wind, (b) Daily average zonal wind. (The black line represents the 0 m/s contour).**

From Fig.1 (a), the meridional wind in this region exhibits significant fluctuations, with frequent alternations in wind direction and large variations in wind speed. The meridional wind shows distinct seasonal variation characteristics: from January to March and from October to December, southerly winds dominate, with occasional fluctuations; from April to September, northerly winds are predominant. From Fig.1 (b), the zonal wind exhibits significant seasonal variation characteristics in both winter and summer, with spring and autumn serving as transition periods. From March to May, the zonal wind gradually decreases with altitude, shifting from westerly to easterly, reaching a maximum easterly wind speed of approximately 22.1 m/s in May. From June to July, the zonal wind is easterly in the 80–85 km range, decreasing with increasing altitude, with a maximum easterly wind speed of 34.2 m/s at 80 km. At approximately 83 km, the zonal wind shifts from easterly to westerly. Between the altitude of wind direction reversal and 100 km, the westerly wind first increases with altitude and then decreases, reaching a maximum westerly wind speed of about 64.4 m/s near 95 km. From September to November, the zonal wind is westerly, with a consistent wind direction throughout the entire altitude range. The westerly wind gradually strengthens with increasing altitude, then starts to weaken above approximately 92 km. At the lower thermosphere, the occurrence and duration of easterly winds increases. In December, January, and February, the zonal wind is westerly. Over time, the westerly wind between 80 km and 90 km gradually strengthens, reaching a maximum westerly




wind speed of approximately 64.3 m/s at 82 km in February. Between 90 km and 100 km, the zonal wind alternates between easterly and westerly.

Figure 2 shows the daily average horizontal wind over Langfang at 1.2–18 km in 2023. The horizontal axis represents time (divided by months), the vertical axis represents altitude (km), and wind speed is measured in m/s. Positive values for the meridional wind indicate southerly wind, and positive values for the zonal wind indicate westerly wind. Blank areas represent missing data.

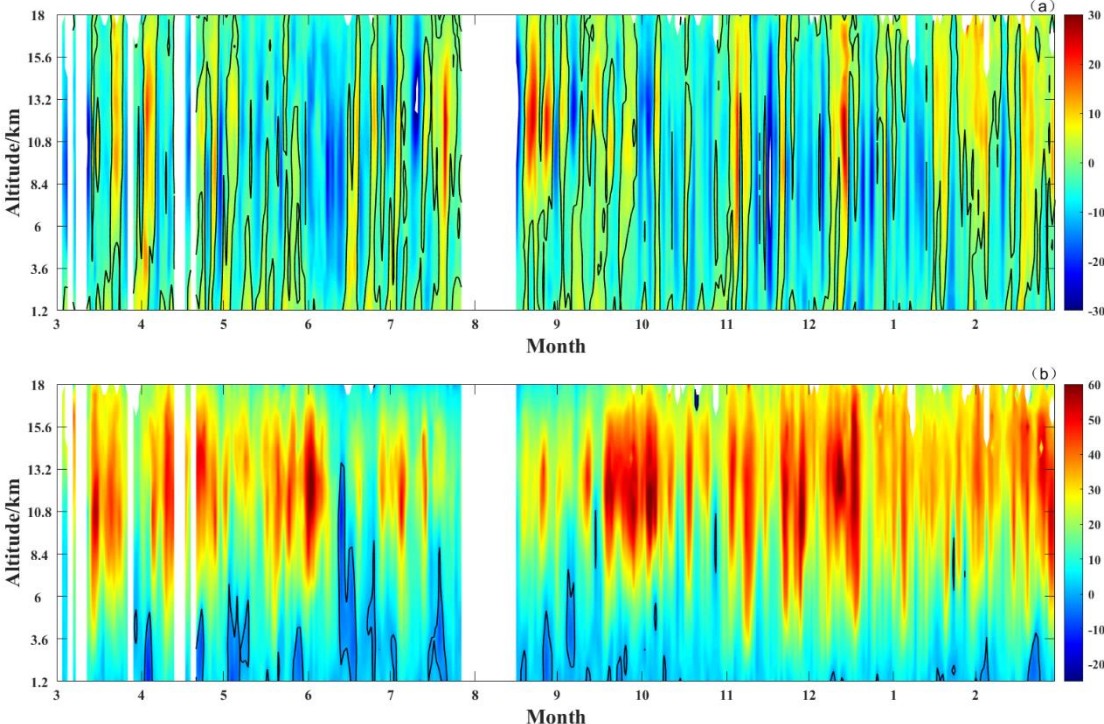


**Figure 2: Daily average horizontal wind at 1.2 – 18 km over Langfang from March 2023 to February 2024: (a) Daily average meridional wind, (b) Daily average zonal wind. (The black line represents the 0 m/s contour).**

From Fig.2 (a), the relationship between the meridional wind and changes in altitude and time can be observed. The meridional wind exhibits complex variations, frequently alternating between southerly and northerly winds. Figure 2 (b)

shows that the zonal wind has higher speeds compared to the meridional wind, with distinct characteristics in winter and summer, while spring and autumn serve as transition periods. From near the surface to 18 km, the zonal wind is westerly, with significant fluctuations. From the surface to around 15 km, the zonal wind gradually increases with altitude, reaching a maximum of approximately 68.4 m/s in the stratosphere between 10 and 15 km. Above 15 km, the zonal wind gradually decreases with altitude. From September to November, the zonal wind is westerly between the surface and 5 km, with

occasional easterly winds. Across the entire altitude range (surface to 18 km), the wind direction remains consistent, with the zonal wind strengthening with altitude and peaking at approximately 68.4 m/s between 8 and 16 km, before weakening



above 16 km. In December, the zonal wind near the surface is westerly, increasing rapidly with altitude, with westerly winds prevailing between 2 km and 18 km. From March to May, the zonal wind alternates between easterly and westerly from the surface to 6 km, with westerly winds being more dominant.

In both Fig.1 and Fig.2, the zonal wind over the MLT and ST above Langfang has higher speeds than the meridional wind. Both the zonal and meridional winds show significant daily variations, indicating that noticeable wind disturbances occur in the MLT and ST over Langfang throughout the year. These prominent disturbances are believed to be related to planetary wave activity (Xiao, Hu, Smith, Xu, & Chen, 2013).

## 3.2 Lomb-Scraggle spectral analysis results

The Lomb-Scraggle spectral analysis was applied to the wind perturbation data, yielding the LS spectral analysis results for the horizontal wind perturbations at different altitudes over Langfang. The tropopause over Langfang is located at approximately 10 km (Xu et al., 2024), so the ST was analysed at 13.8 km and 7.2 km to observe results in the stratosphere and troposphere. The MLT was analysed at 96 km and 84 km to observe results in the lower thermosphere and mesosphere. Figure 3 shows the spectral analysis results for these four altitudes. The horizontal axis represents time (divided by months),

the vertical axis represents frequency ($day^{-1}$), the intensity is in m/s, and blank areas indicate missing data.

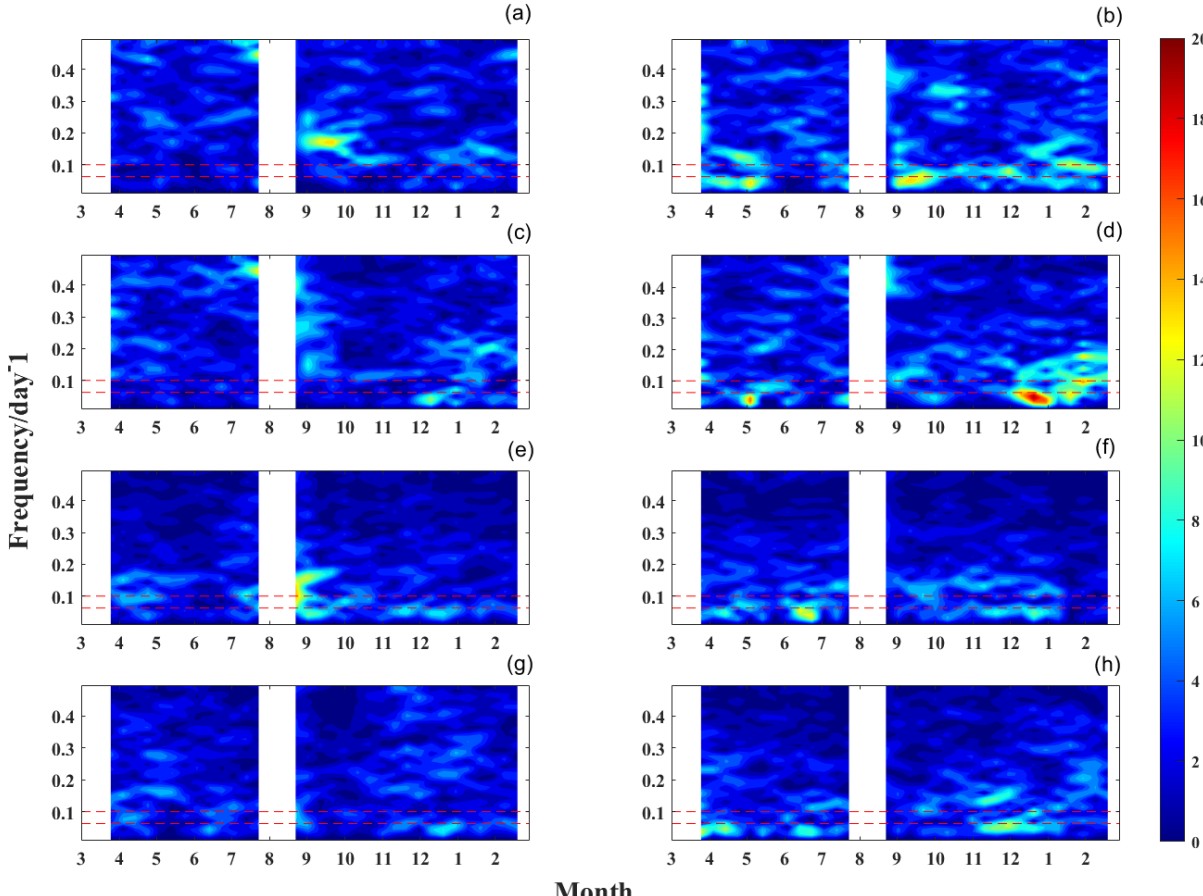

**Figure 3**: **LS spectral analysis results of horizontal wind perturbations at different altitudes over Langfang from March 2023 to February 2024: (a), (c), (e), (g) are the results of the LS spectrum analysis of the meridional wind disturbance at 96 km, 84 km, 13.8 km, and 7.2 km, respectively; (b), (d), (f), (h) are the results of the LS spectrum analysis of the zonal wind disturbance at 96 km, 84 km, 13.8 km, and 7.2 km, respectively. (The red dashed lines from top to bottom correspond to the frequencies of 10-day and 16-day planetary waves.)**

As shown in Fig.3, the frequency distribution of meridional and zonal wind perturbations at the four altitudes is quite extensive. The LS analysis results for the meridional wind on the left show that quasi-16-day, quasi-10-day, and quasi-5-day planetary wave perturbations are prominent at 7.2 km and 13.8 km. Additionally, noticeable quasi-5-day planetary wave perturbations appear at 13.8 km, 84 km, and 96 km in September. Besides the quasi-16-day, quasi-10-day, and quasi-5-day waves, the 96 km and 84 km altitudes also exhibit a strong quasi-2-day component, which intensifies rapidly at these altitudes in July. Due to missing data in August, further development of this component cannot be assessed. The LS analysis results for the zonal wind on the right show that quasi-16-day and quasi-10-day planetary wave perturbations dominate at all





four altitudes, with perturbation intensity generally stronger at 84 km and 96 km compared to 7.2 km and 13.8 km, and with

the strongest perturbations occurring at 84 km.

By combining the LS analysis results for both meridional and zonal winds at the four altitudes, it is evident that zonal wind perturbations have stronger amplitudes than meridional wind perturbations. In terms of temporal distribution, the quasi-16-day wave dominates for a longer duration compared to the quasi-10-day wave. This time distribution and amplitude variation trend is consistent with the LS spectral analysis of zonal wind at 84 km using data from the Fengcheng meteor radar (Yang et

al., 2023).From Fig.3 (b) and (d), in April, there are strong quasi-16-day and quasi-10-day planetary wave perturbations at 96 km, while the quasi-16-day and quasi-10-day planetary wave perturbations at 84 km are weak. In September, the quasi-16-day planetary wave perturbations at 96 km are also stronger than those at 84 km, whereas in December, the quasi-16-day planetary wave perturbations at 84 km are stronger than those at 96 km.

From Fig.3 (f) and (h), in April, June, November, and December, both the quasi-16-day and quasi-10-day waves are clearly

present simultaneously in the troposphere and stratosphere. However, Figure 3 (e) and (g) show that in June, the quasi-16-day planetary wave perturbations exhibit strong activity only in the troposphere, while Fig.3 (d) and (f) indicate that in May, the quasi-10-day planetary wave perturbations exhibit strong activity only in the stratosphere. This suggests that the planetary waves observed in the stratosphere may have propagated upward from the troposphere or could have propagated meridionally from the stratosphere at higher latitudes, which is consistent with observations made by Tang Lei et al. using

the Xianghe MST radar (Lei Tang 2016).

This study primarily focuses on the quasi-16-day planetary wave. To analyse its amplitude variation with height and time, as well as the phase variation with height and time, we further applied band-pass filtering and harmonic fitting methods to analyse the wind perturbation field in greater detail.

## 3.3 Spatiotemporal variations of the 16-day period planetary wave

A band-pass filter with a range of 12–20 days was applied to the wind perturbations, and the results reflect the variation of the quasi-16-day wave with height and time. The band-pass filtering results are shown in Fig.4. Both the MLT and ST exhibit quasi-16-day planetary wave perturbations, with the maximum intensity of the quasi-16-day wave being higher in the MLT than in the ST. Additionally, the quasi-16-day wave perturbations are stronger in the zonal wind than in the meridional wind. No significant planetary wave activity is observed in the MLT during summer, while noticeable planetary wave

activity is present in the ST during this period.





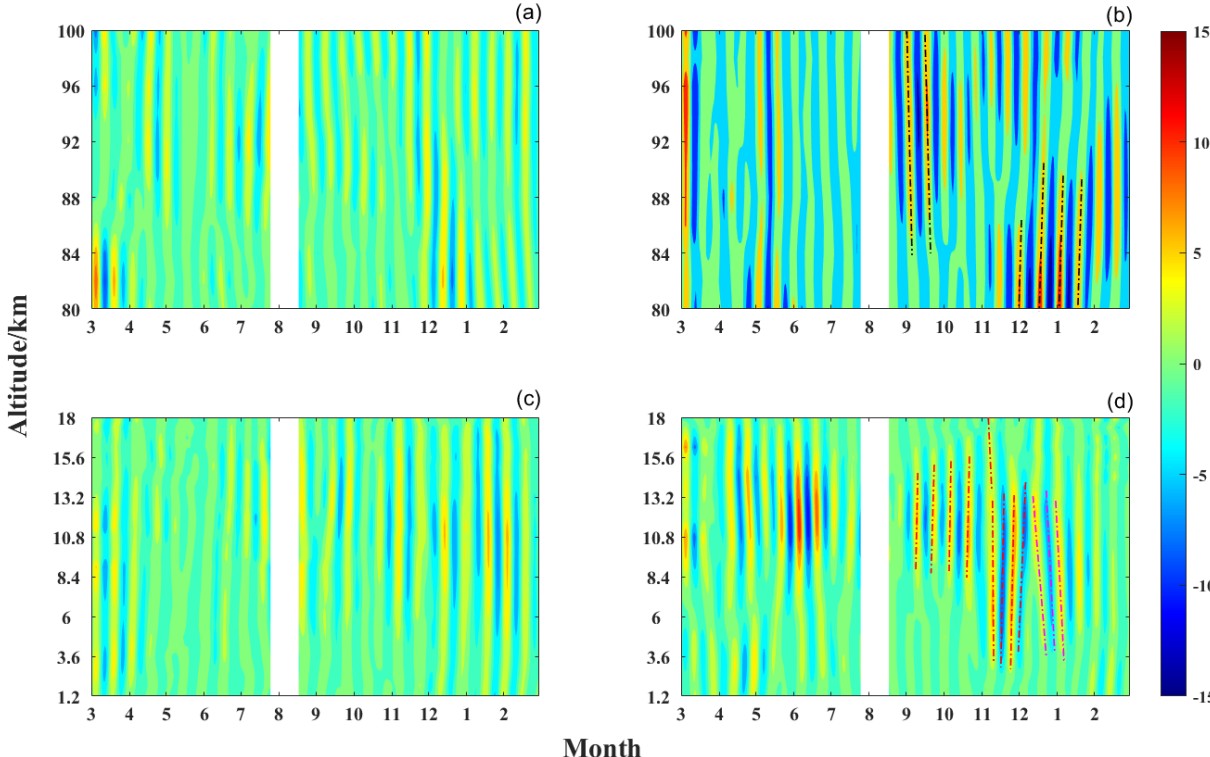

**Figure 4**: **Structure of daily average horizontal wind perturbations after applying a 12–20 day band-pass filter: (a) Structure of daily average meridional wind perturbations at 80–100 km altitude; (b) Structure of daily average zonal wind perturbations at 80–100 km altitude; (c) Structure of daily average meridional wind perturbations at 1.2–18 km altitude; (d) Structure of daily average zonal wind perturbations at 1.2–18 km altitude. (Blank areas indicate periods where data coverage is less than 50%, and the dashed lines represent the slope of the phase fronts.)**


In the MLT meridional wind, strong quasi-16-day planetary wave perturbations appear in winter and spring, occurring in the 80–90 km and 80–85 km altitude ranges, respectively, with a maximum amplitude of 14 m/s observed in March. In the zonal wind, the quasi-16-day planetary wave perturbations are weak during summer, with amplitudes mostly below 5 m/s. These

patterns are consistent with previous findings (Day & Mitchell, 2010; J. M. Forbes et al., 1995; Gong et al., 2020; Yang et al., 2023). These perturbations are more prominent in May and during the autumn and winter seasons. In May, they nearly span the entire 80–100 km altitude range, while in autumn they are concentrated between 85 and 100 km. In winter, they are divided into two sections, at 80–88 km and 92–100 km, and by mid-December, they nearly cover the entire 80–100 km range. The maximum amplitude of 15 m/s occurs in December, which aligns with the timing of the maximum amplitude of the

quasi-16-day wave in this region observed at Fengcheng (Yang et al., 2023). The filtering results for the MLT are consistent with the time distribution of the quasi-16-day wave filtering results obtained by Huang, XS et al. using data from the Beijing meteor radar (X. S. Huang et al., 2022), with stronger activity observed in autumn and winter. There are slight differences in altitude distribution, as the results in September and November from this study do not cover the entire 80–100 km range.



This difference may be due to the quasi-16-day planetary wave source in September being located above 100 km, or the

strong 16-day wave being propagated meridionally from high latitudes. The discontinuity at 85–90 km in November may

indicate that the quasi-16-day planetary wave around 90 km was unable to continue propagating upward. Using harmonic

fitting, the phase information of the 16-day wave within the 88–100 km range in September was obtained. A linear fit using

the least squares method yielded a vertical wavelength of 148 km for the 16-day planetary wave at this altitude during this

period. Using the same method, the vertical wavelength of the 16-day planetary wave in the 80–85 km range from December

to February was determined to be 112 km.

The temporal and spatial distribution of the meridional and zonal components of the 16-day wave in the ST is broader and

more uniform. In the ST, strong quasi-16-day planetary wave perturbations in the meridional wind are present continuously,

except in May, with the strongest activity occurring in winter. These perturbations are distributed between 1.2 km and 18 km,

with higher intensity in the 5–15 km range, consistent with the findings of Lu Xian et al.'s study of quasi-16-day planetary

waves in this region using radiosonde data (LU Xian, 2005). The quasi-16-day planetary wave perturbations in the zonal

wind are stronger in spring, summer, and autumn. In spring and summer, they are distributed between 9 km and 18 km,

while in autumn, they are primarily found between 4 km and 15 km. From a temporal perspective, planetary wave

perturbations in the ST occur frequently but intermittently, with durations of 1 to 3 months, and there is no clear pattern to

their occurrence (Lei Tang 2016). The vertical wavelength of the 16-day planetary wave between 8.5 km and 15 km in

September and October is 100 km. In November and January, the vertical wavelength of the 16-day planetary wave is 119

km and 109 km, respectively, within the 5–13 km range.

It is noteworthy that the phase propagation direction of planetary waves in both regions reversed during November to

December, as indicated by the dashed lines in Fig.4 (b) and (d). From September to November, the phase of the planetary

waves in the MLT propagated downward, while in the ST, the phase of the planetary waves in the 10–16 km altitude range

propagated upward. However, from mid-December to January of the following year, the phase of the planetary waves in the

MLT propagated upward, with the phase of the planetary waves in the ST propagating upward in the 13–18 km altitude

range and downward below 13 km. A similar reversal phenomenon in the MLT was also observed by Huang, XS in their

study of anomalous quasi-16-day wave activity from October 2013 to January 2014 using meteor radar in mid-latitude

regions (X. S. Huang et al., 2022). This reversal was evident in the quasi-16-day zonal component data from both the Wuhan

meteor radar and the Beijing meteor radar from December to January of the following year, although the phenomenon was

not explained in that study.

The reversal in the phase propagation direction mentioned above may be caused by the relationship between the phase speed

and group velocity during the vertical propagation of planetary waves. A three-dimensional planetary wave can be expressed

as $\exp\left(i(\omega t - kx - ly - mz)\right)$, and its vertical group velocity can be expressed as:

$c_g = \partial\omega/\partial m = \partial\bar{q}/\partial y \cdot 2mk/(k^2 + l^2 + m^2)^2$ (In a stable atmosphere,$\partial\bar{q}/\partial y > 0$), (3)





The phase speed is $c_p = \omega/m$, so when $k < 0$, the vertical group velocity is opposite to the phase speed, and when $k > 0$, the vertical group velocity is in the same direction as the phase speed.

To explore the cause of this phenomenon, the frequency-wavenumber spectra was obtained through a two-dimensional Fourier transform of the detrended zonal wind perturbations at 10 km and 79 km altitudes, using MERRA-2 reanalysis data
from August 28 to September 28, November 12 to December 13, and December 18 to January 18. The results are shown in Fig.5. Assuming that the frequency-wavenumber spectrum in the MLT is consistent with that at the 79 km altitude, the dominant wavenumber for each time was selected as the wavenumber for that period. The relationship between the wavenumbers at the two altitudes, the phase front trends, and the energy propagation direction for the three time periods is summarized in Table 3.

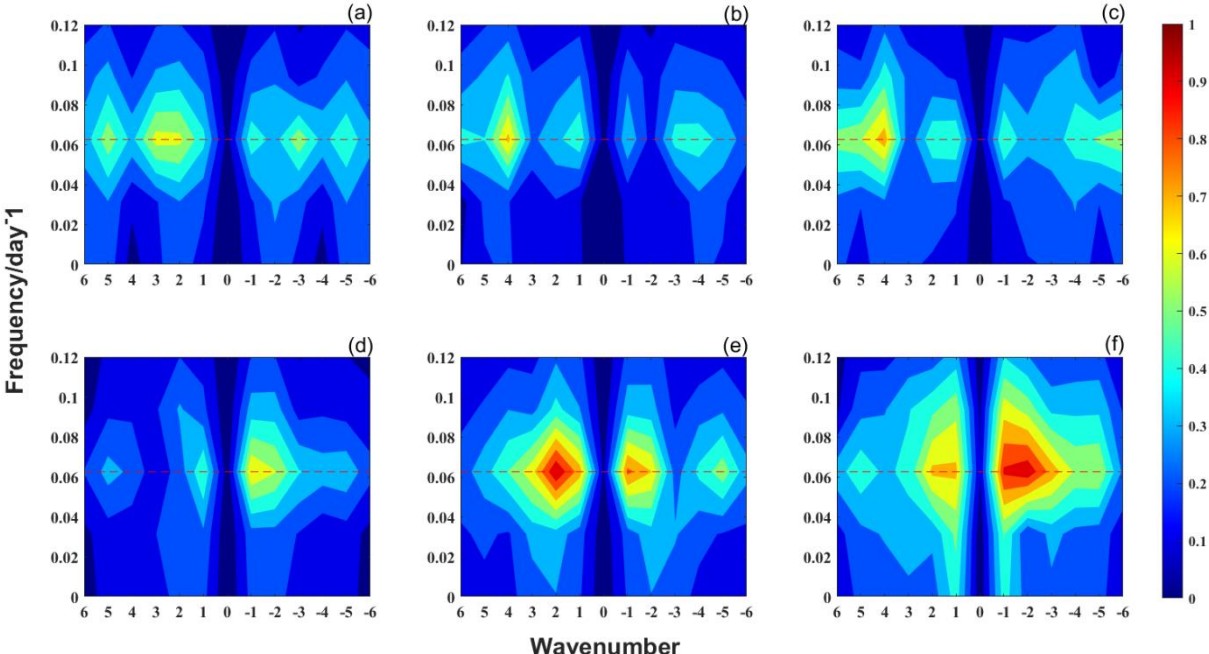


**Figure 5**: **Frequency-wavenumber spectra at 79 km and 10 km altitudes: (a), (b) and (c) are the frequency-wavenumber spectra at 79 km for the periods of August 28 to September 28, November 12 to December 13, and December 18 to January 18, respectively; (d), (e) and (f) are the frequency-wavenumber spectra at 10 km for the same periods. (The intensity is the normalized result. The red dashed lines correspond to a 16-day period, and negative wavenumbers indicate westward propagation.)**

**Table 3 Relationship between wavenumbers at two altitudes, phase fronts, and energy propagation direction during the three time periods.**

| Altitude | Parameters | Aug.28 – Sept.28 | Nov.12 – Dec. 13 | Dec.18 –Jan.18 |
|---|---|---|---|---|
| | **Wavenumber** | +2/+3 | +4 | +4 |
| **79km** | **Direction of $c_g$ vs $c_p$** | Same | Same | Same |
| | **Trend of phase front changes** | Downward | Upward | Upward |





| | | | | |
|---|---|---|---|---|
| **10km** | **Energy propagation direction** | Downward | Upward | Upward |
| | **Wavenumber** | -1 | 2 | -1/-2 |
| | **Direction of $c_g$ vs $c_p$** | Opposite | Same | Opposite |
| | **Trend of phase front changes** | Upward | Upward | Downward |
| | **Energy propagation direction** | Downward | Upward | Upward |

Combining Fig.4 (b), (d) and Table 3, during the periods from November 12 to December 13 and December 18 to January 18, the energy of the 16-day planetary wave propagated upward within the 3–13 km range. However, the phase front trends with height and time were opposite during these two periods, due to a reversal in the horizontal direction of the 16-day wave. This is consistent with the theory regarding the influence of positive or negative wavenumbers on the direction of vertical group velocity and phase velocity. Additionally, the wave modes in the two regions from August 28 to September 28 differ, indicating that the planetary waves in the two regions do not share the same origin. The energy of the 16-day planetary wave in the 85–100 km range propagates downward, suggesting that the source of the planetary wave is at a higher altitude or that the wave propagated meridionally from high latitudes. In the 8–14 km range, the energy of the 16-day planetary wave also propagates downward, while in the 14–18 km range, the energy propagates upward, indicating that the source of the planetary wave may be near 14 km. From November 12 to December 13 and December 18 to January 18, the energy of the 16-day planetary wave in the 80–85 km range propagates upward, and the energy of the 16-day planetary wave in the 3–13 km range also propagates upward. Although the dominant wave modes in the two regions are different, the secondary wave mode in the MLT partially overlaps with the dominant wave mode in the ST. This suggests that the planetary wave in the MLT may have propagated vertically upward from the ST.

## 3.4 Propagation characteristics of the 16-day period planetary wave

To investigate the location of the wave source in the ST from August 28 to September 28, as well as the relationship between the planetary waves in the two regions from November 12 to December 13 and December 18 to January 18, harmonic fitting was applied to MERRA-2 reanalysis data and meteor radar data. After removing the quasi-16-day fluctuations, the zonal background wind from near the surface to 79 km and from 80 to 100 km was obtained for the period from August 28 to January 31, as shown in Fig.6. Blank areas represent data gaps caused by the FFT window. To study the vertical propagation conditions over Langfang, the zonal background wind from near the surface to 79 km is used as an approximation for the zonal background wind at the same latitude.

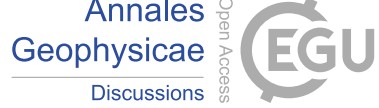

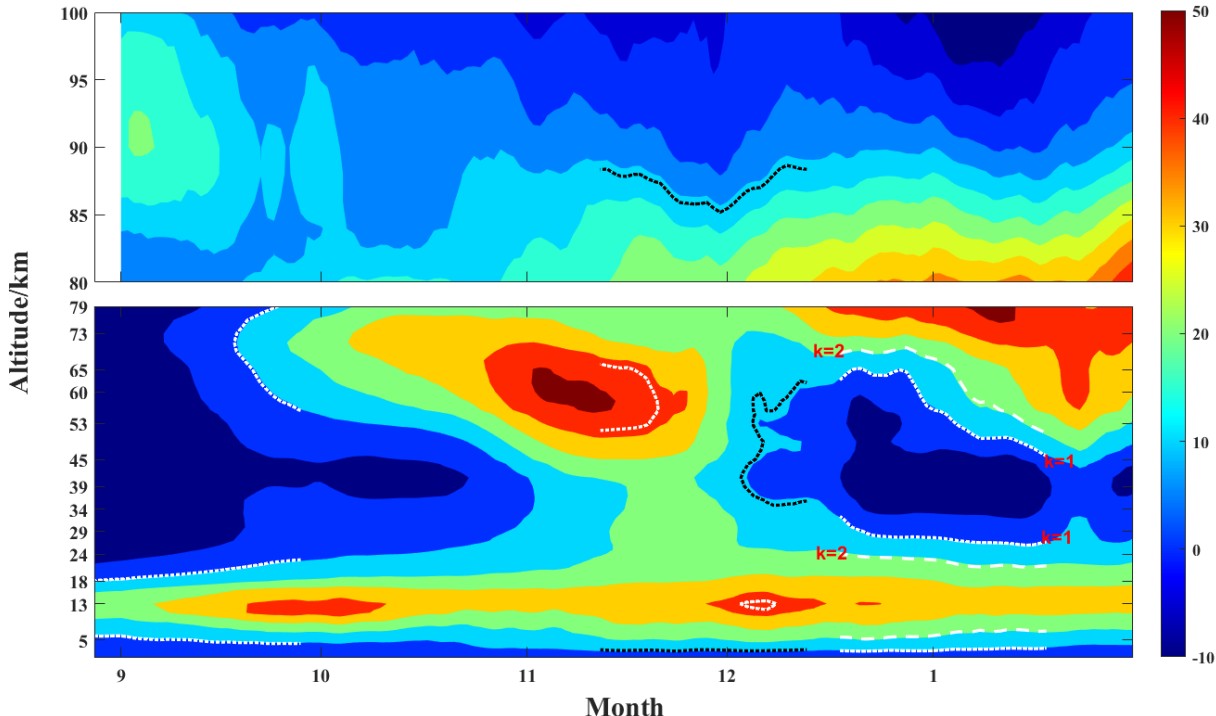

**Figure 6**: **Zonal background wind in the altitude ranges from near the surface to 79 km, and from 80 to 100 km (the black line represents the lower limit contour for vertical propagation conditions, the white line represents the upper limit contour for vertical propagation conditions).**

From Fig.6, between August 28 and September 28, 2023—particularly from September 5 to September 28—the zonal background wind near 13 km is exceptionally strong. This altitude typically corresponds to the jet stream region, where the strong horizontal wind speed gradient is a key factor in the formation of planetary waves. Variations in wind speed above and below the jet stream can cause instabilities, often triggering planetary waves. This verifies the hypothesis from section 2.3 that the planetary wave source in the ST from August 28 to September 28 may be located near 14 km. During the period from August 28 to September 28, the wavenumber $k = 1$, and the planetary wave propagated westward. The horizontal wavelength $\lambda = (2\pi R \cos \phi)/k$, where $R$ is the Earth's radius (approximately 6371 km) and $\phi$ is the latitude, gives $\lambda$ of approximately 30,943 km. The corresponding phase velocity is -22.4 m/s (positive eastward). According to the classical theory of planetary wave propagation, when the condition $0 < \bar{u} - c_x < U_c$ is satisfied, the planetary wave will propagate vertically from the troposphere to the stratosphere and mesosphere. Assuming the meridional wavenumber $l$ of the 16-day wave is 3 (the same assumption is used hereafter), based on equation (2), we obtain $U_c \approx 32 m/s$. For the planetary wave to propagate vertically, the average zonal wind $\bar{u}$ should satisfy $-22.4 m/s < \bar{u} < 9.6 m/s$. From Fig.6, the average zonal wind around 18 km during this period does not meet the conditions for vertical propagation of the planetary wave. Figure 4 (d) also shows that the 16-day planetary wave activity around 18 km was quiet during this period.





During the period from November 12 to December 13, the wavenumber $k = 2$, and the planetary wave propagated eastward, giving a horizontal wavelength $\lambda$ of approximately 15,472 km. The corresponding phase velocity is 11.2 m/s (positive eastward). With $U_c \approx 32m/s$ , for the planetary wave to propagate vertically, the average zonal wind $\bar{u}$ must satisfy

$11.2m/s < \bar{u} < 43.2m/s$. From Fig.6, during this period, the conditions for vertical propagation of the planetary wave are satisfied, especially from November 21 to December 3, where the entire altitude range from near the surface to 87 km meets the conditions for vertical propagation. During this time, it is possible that the planetary wave in the MLT propagated vertically upward from the ST.

During the period from December 18 to January 18, the dominant wavenumbers were $k = 1$ and $k = 2$, with the planetary

wave propagating westward. For $k = 1$, the horizontal wavelength $\lambda$ is approximately 30,943 km, corresponding to a phase velocity of -22.4 m/s (positive eastward). The condition for planetary wave vertical propagation is $-22.4m/s < \bar{u} < 9.6m/s$. For $k = 2$, the horizontal wavelength $\lambda$ is approximately 15,472 km, corresponding to a phase velocity of -11.2 m/s (positive eastward). The condition for planetary wave vertical propagation is $-11.2m/s < \bar{u} < 20.8m/s$. From Fig.6, during this period, the conditions for vertical propagation of planetary waves to the MLT are not met. Therefore, the

planetary waves in the MLT during this period are not caused by upward propagation from the ST. They may have been triggered by the planetary waves propagating upward from the westerly jet near 73 km, or they may be a continuation of the planetary waves that propagated upward between November 12 and December 13, continuing their activity from December 18 to January 18.

## 4 Conclusion

Using horizontal wind data from the dual-frequency ST-M radar at the Langfang Observatory from March 2023 to February 2024, this study investigated the daily average horizontal wind structure and planetary wave characteristics in the troposphere and stratosphere, as well as in the mesosphere-lower thermosphere over Langfang. Special attention was given to the spatiotemporal variations, propagation characteristics of the quasi-16-day wave, and the relationship between planetary waves in these two regions. The conclusions are as follows:

(1) Over Langfang, the daily average zonal wind in both the MLT and ST is stronger than the meridional wind, and both the zonal and meridional winds exhibit significant daily variations. This indicates that there are noticeable wind disturbances in the MLT and ST over Langfang throughout the year. These disturbances are considered to be related to planetary wave activity.

(2) Through LS spectral analysis, it was found that there are planetary waves activities with periods of 2-20 days all the year.

The quasi-16-day and quasi-10-day waves dominate in both the ST and MLT regions, with the quasi-16-day wave being more prominent. In April, October, November, and December 2023, both quasi-16-day and quasi-10-day waves were present in the troposphere and stratosphere, while in May, only the quasi-10-day wave was observed in the stratosphere.





(3) Analysis of the filtering results shows that quasi-16-day planetary wave activities occur throughout the year, its zonal component is stronger than the meridional component, and it is characterized by being strong in winter and weak in summer.

Notably, we discovered that the direction of vertical phase propagation for the quasi-16-day waves changes with the seasons. During autumn (August-September), these waves propagate upward in the ST and downward in the MLT. In November-December, they continue to move upward in the ST while also propagating upward in the MLT. However, after December, the waves move downward in the ST but still upward in the MLT. We observed that the dominant wavenumbers for the 16-day waves are as follows: in August-September, they are (-1 for ST and 2 for MLT); in November-December, they shift to (2

for ST and 4 for MLT); and in December-January, they return to (-1 for ST and 4 for MLT). The direction of vertical phase propagation of the 16-day wave varies with the seasons due to the reversal of the horizontal propagation direction of the planetary waves. When the planetary waves propagate westward, the direction of energy propagation is opposite to the direction of phase propagation, whereas when the planetary waves propagate eastward, the direction of energy propagation aligns with the direction of phase propagation.

(4) By analysing the vertical phase velocity, zonal wavenumber, and vertical propagation conditions, we determined that the group velocity of the 16-day waves is downward in both ST and MLT during August-September. Conversely, it is upward in both layers in November-December, and again downward in ST but upward in MLT in December-January. We infer that the observed 16-day waves in the ST may be initiated by jet streams around 14 km altitude, allowing them to propagate downward in August-September. However, background winds prevent these waves from moving upward. The waves

observed in the MLT during this time may originate from unknown sources or may be refracted from higher or lower latitude regions. The behaviour of the 16-day waves in ST changes in November-December, indicating they are generated in the lower atmosphere and can propagate upward through the background winds into the MLT. In December-January, the waves in ST again exhibit different wavenumbers and continue to propagate upward, though their ascent is likely blocked by upper-level winds, preventing penetration into the MLT. The observed MLT waves in December-January are likely remnants of

previously existing waves.

**Data availability**

MERRA-2 (Modern-Era Retrospective Analysis for Research and Applications, Version 2) is a state-of-the-art atmospheric reanalysis dataset developed by NASA's Goddard Space Flight Center (GSFC). It provides high-resolution reconstructions of atmospheric conditions, integrating satellite observations, ground-based measurements, and numerical modeling, accessible

at https://disc.gsfc.nasa.gov/datasets/M2I3NVASM_5.12.4/summary?keywords=MERRA (M2I3NVASM).

The ST-M radar data are not publicly available due to the laboratory policies or confidentiality agreements.



## Author contributions

Conceptualization, ZZ, XH and QX; methodology, ZZ, XH, QX, BC and JY; formal analysis, ZZ, BC and JY; validation, ZZ, XH and BC; resources, QX.; data curation, ZZ and QX; writing—original draft preparation, Z.Z.; writing—review and

editing, ZZ, XH, BC and JY; supervision XH and QX; funding acquisition, QX and JY. All authors have read and agreed to the published version of the manuscript.

## Competing interests

The authors declare no conflicts of interest.

## Acknowledgments

The authors wish to express their sincere thanks to the NASA Goddard Earth Sciences (GES) Data and Information Services Centre (DISC) for providing MERRA-2 reanalysis data (GES DISC Dataset: MERRA-2 inst3_3d_asm_Nv: 3d,3-Hourly,Instantaneous,Model-Level,Assimilation,Assimilated Meteorological Fields V5.12.4 (M2I3NVASM 5.12.4) (nasa.gov) , last access: 20 November 2024).

## Financial support

This research was funded by the National Natural Science Foundation of China (Grantnumbers 12241101, 42174192, and 11872128), and the "Climbing Plan" of the National Space Science Center, Chinese Academy of Sciences. The APC was funded by the National Space Science Center, Chinese Academy of Sciences.

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
