# Peer review of "Research on 16-day Planetary Waves in the Mid-latitude Troposphere, Stratosphere, Mesosphere, and Lower Thermosphere with Langfang Dual-frequency ST-M Radar Data"

_Annales Geophysicae, 2024_

## Referee Comment (RC2)

Comments on the paper "Research on 16-day Planetary Waves in the Mid-latitude Troposphere, Stratosphere, Mesosphere, and Lower Thermosphere with Langfang Dual-frequency ST-M Radar Data" by
Zengmao Zhang, Xiong Hu, Qingchen Xu, Bing Cai, Junfeng Yang

The paper presents new data about the wind variability in the MLT and in the troposphere/lower stratosphere, as well as some results on the coupling between these atmospheric layers. The methods used in the analysis are not well justified and are sometimes incorrect. Therefore the paper needs a major revision. The detail comments are given below.

**The title** does not clearly describe the article. The radar data cover the lower stratosphere and the MERRA data are not mentioned.

**Data Sources and Analysis Methods**

1. Extracting Daily Mean Wind:
Why do the authors use the criterion of 10 hours? What are the errors of the daily wind speeds obtained with the method proposed by the authors?
2. Detrending
The MLT winds are characterized by the strong seasonal course and strong changes in spring, sometime in autumn and during SSW (for example, in January 2024). The wind behavior model with a simple linear trend is not correct for these cases. Therefore, the 16-day wave parameters may be obtained with large errors or may be completely incorrect. I recommend removing the seasonal course first.
3. Spectral analysis
The time series of the zonal and meridional wind speeds have large gaps from month 3 to month 5 (figure 1). On the one hand, such gaps may easily distort the spectrum obtained with the Lomb-Scargle method. On the other hand, the authors fill in the gaps for their further analysis.
I recommend removing the data with these large gaps.

Figure 3.  Please, indicate units of the color levels. What level is significant?

**3.3 Spatiotemporal variations of the 16-day period planetary wave**
L.325-330  "No significant planetary wave activity is observed in the MLT during summer…"
How do the authors separate significant and non-significant wave activity?

Ln. 350-365 Please, indicate errors of the vertical wavelength you found.

Ln.367-376 It is very important for the analysis provided in this part and below that the errors of the phase speeds are small enough to draw any conclusion.

Ln.380   $Q_y < 0$ is not sufficient for the instability.

Ln.380-383  The authors use the result of the quasi-geostrophic theory. The conclusion from eq.3 is not correct.  The vertical phase speed is opposite to the vertical group speed in the coordinate system that moves with the zonal flow ($Q_y > 0$). If one takes into account the background zonal flow $U_0$, then the result will be complex and will depend on $U_0$.
By the way, the authors should explain the notation in the equation and provide a reference.

L.385 "Fig.5. Assuming that the frequency-wavenumber spectrum in the MLT is consistent with that at the 79 km altitude, the dominant wavenumber for each time was selected as the wavenumber for that period. "

Why are the spectra consistent? The MERRA-2 data are given at the model level, but the MR winds are given at altitudes. The difference between the true heights may be significant.

There is no confidence that oscillations presented in Fig.5 are statistically significant.

**3.4 Propagation characteristics of the 16-day period planetary wave**

The aim of this part is "To investigate the location of the wave source in the ST" and "the relationship between the planetary waves in the two regions".

The analysis is confined in latitude and longitude to the region where the radar is located. Therefore, the authors (and the readers) do not know how 16-day waves propagate in the neighboring region. Hence, the authors can't really reach their aim.

Additional note, the real atmospheric 16-day waves are transient, their amplitudes are changing with time as observed. The theory used in this part does not work for such waves.

I propose to find and plot the E-P flux for the waves.

**Conclusion**

Ln.465 "The quasi-16-day and quasi-10-day waves dominate in both the ST and MLT regions" – this conclusion may be a result of the 32-day segment used for the analysis and a linear trend model. The waves with shorter periods are just averaged over the segment and their transient behavior is not taken into account.

Conclusions (2) and (3) repeat the first one. The errors of phase speeds are not clear. Therefore, the statements about their changes are not supported in the text.

Ln.475-479 Please, see above. The authors' statements are incorrect.

Ln. 480 Conclusion (4). This conclusion does not have a solid support from the analysis as it is noted above. The sign of speeds, the wavenumber estimations are in question.

---

## Author Comment (AC1)

**Replies to the Reviewer#1**

**Dear Reviewer,**

Thank you very much for your feedback and constructive comments. I sincerely appreciate the time and effort you devoted to reviewing my manuscript. Below are my responses to the issues you raised. In this reply, the comments of the referee are marked in black color, and the replies in blue color.

**Comments 1**: Lack of Novelty or Contribution

Reviewer Comment: "Similar observations of planetary waves using ST radar and meteor radar at mid-latitudes already exist. The study does not present new knowledge about the conditions for vertical planetary wave propagation."

**Reply 1**:

Thank you again.

We believe that the manuscript presents innovative results, as explained below:

**(1)** Based on our analysis, we have, for the first time, identified a reversal in the vertical propagation direction of the 16-day planetary wave phase in the ST region during the transition from autumn to winter (November–December), as shown in Figure 4(d) of the manuscript.

**(2)** By incorporating reanalysis data, we provide a reasonable explanation for this phenomenon. We identify that the horizontal wave mode of the 16-day planetary wave undergoes a change, resulting in a reversal of its horizontal propagation direction. This reversal subsequently leads to a change in the direction of vertical phase propagation. However, the direction of vertical group velocity remains unchanged, consistently propagating upward. This finding differs from the previously reported understanding in the literature(Jiang, Xiong, Wan, Ning, & Liu, 2005; Lu X & Zhang, 2005; Y. Luo, Manson, Meek, Thayaparan, et al., 2002; Tang & Huang, 2016), which suggested that the vertical group velocity and vertical phase velocity of the 16-day planetary wave propagate in opposite directions.

**(3)** Based on the significant 16-day wave observed by the ST-M radar from September to the following January, along with MERRA-2 reanalysis data, we discussed the relationship between the 16-day planetary waves in the MLT and ST regions above Langfang Station: the significant 16-day wave observed at 90–100 km during September 3–18 may have originated from meridional propagation from other latitudinal regions; the 16-day wave in the ST region during November 27–December 2 and December 6–23 may have propagated vertically into the MLT region; the significant 16-day wave observed at 80–85 km in the MLT region during December 27–January 14 may have originated from meridional propagation from other latitudes, or it may be a continuation of the upward-propagating 16-day wave from the December 6–23 period.

The above findings enhance our understanding of the 16-day planetary wave.

We have made corresponding revisions to the manuscript by adding this content to the Introduction and Conclusion sections to clearly highlight the innovative aspects of the study.

45

**Comments 2:** Poor Writing That Hampers Understanding

Reviewer Comment: "The manuscript is difficult to read due to numerous spelling errors and unusual or misleading word choices. Many statements lack clarity."

**Reply 2:**

50    We have carefully revised the entire manuscript. We used tools such as DeepL Translator and the Microsoft Word editor to check the spelling throughout the text. We also standardized the labels in the figures and revised some long sentences to improve readability.

55    **Comments 3:** Methodological Concerns and Suggestions

Reviewer Comment: The authors aim to estimate the period and vertical wavelength (hence vertical phase speed) of observed planetary waves. However, the reviewer finds the approach inefficient.

**Reply 3:**

60    Thank you!

The radar data in the manuscript were processed using the following methods:

• **Lomb–Scargle (LS) Spectral Analysis**

• **Band-pass Filtering**

• **Harmonic Fitting**

65    The purpose of using these methods is clearly defined. Below, we provide explanations for each method used:

The LS method was applied using a 32-day sliding time window to obtain the variation of planetary wave amplitudes with different periods at four altitudes, in order to assess the relative importance of 16-day planetary wave activity throughout the year. In fact, due to

70    the limited amount of missing data, it is also possible to apply interpolation followed by Fourier transform as an alternative approach.

For the analysis of the spatiotemporal distribution of planetary waves and the estimation of vertical phase and vertical wavelength, we did not use the LS method. Instead, we applied

75    **band-pass filtering** and **harmonic fitting**.

• **Band-pass filtering** was used to extract the temporal variation characteristics of the 16-day wave. This method has been widely adopted in previous studies to characterize the

80    temporal evolution of 16-day planetary waves(Day & Mitchell, 2010; Huang et al., 2022; G. Jiang et al., 2005; G. Y. Jiang et al., 2005; Lima, Batista, Clemesha, & Takahashi, 2006; Y. Luo, Manson, Meek, Meyer, et al., 2002; Yi Luo, Manson, Meek, Meyer, & Forbes, 2000; Namboothiri, Kishore, & Igarashi, 2002; Tang & Huang, 2016; Vineeth, Pant, Kumar, Ramkumar, & Sridharan, 2009). The reason for using the same method in this study is that it

85    provides clear and intuitive temporal variation characteristics, which facilitate comparison with results from previous studies.

• **Harmonic fitting** (see Equation (1) in the manuscript) was applied to estimate the vertical

phase slope and the zonal background wind of the 16-day planetary wave. The zonal wind
was further used to evaluate the vertical propagation conditions of the 16-day wave and to
analyze the relationship between wave activity in the ST and MLT regions.

(1) A 32-day sliding window was used to perform harmonic fitting of the 16-day planetary
wave, yielding the amplitude and phase at different altitudes.
(2) The altitude range with relatively large 16-day wave amplitudes was identified, and the
positions of wave peaks at different heights were extracted. A linear fit of the form $\varphi(z) =
mz + \varphi_0$ (where z is altitude and $\varphi_0$ is the initial phase) was then applied to obtain the
vertical phase slope.
(3) The sign of vertical phase slope was used to determine the direction of the apparent
vertical phase speed of the planetary wave ($m > 0$ indicates upward propagation, while $m <
0$ indicates downward propagation), and the vertical wavelength was calculated using $\lambda =
2\pi/m$.

**Comments 3.1:** Step 1: This step is valid and reasonable. However, the computation of the
error of daily mean winds and wind variances should be included. These can be used to
establish significance levels in the Lomb-Scargle (LS) analysis.
**Reply 3.1:**
Adopted.
We calculated the errors of the daily mean wind speed and wind variance and added
information on the confidence level in the LS analysis section and updated Figure 3 in the
manuscript, where the white dashed lines represent the 90% confidence level.

[Figure]

Figure 3: LS spectral analysis results of horizontal wind perturbations at different altitudes
over Langfang from March 2023 to February 2024: (a)(c)(e)(g) are the results of the LS
spectrum analysis of the meridional wind disturbance at 96 km, 84 km, 13.8 km, and 7.2 km,

respectively; (b)(d)(f)(h) are the results of the LS spectrum analysis of the zonal wind disturbance at 96 km, 84 km, 13.8 km, and 7.2 km, respectively. (The red dashed lines from top to bottom correspond to the frequencies of 10-day and 16-day planetary waves. The white dotted line represents 90% confidence level.)

120

**Comments 3.2:** Step 2: This step is unnecessary.

**Reply 3.2:**

The main purpose of detrending is to remove background trends in the data (such as seasonal increases or decreases), in order to better analyze the underlying fluctuations.

125 Therefore, we have decided to retain this step.

**Comments 3.3:** Step 3:

- ○ The choice of Lomb-Scargle spectral analysis is appropriate given the unevenly sampled data. This method provides spectral amplitudes and

130 phases as functions of both frequency and time/altitude thanks to the usage of a sliding window.

- ○ However, a proper significance analysis, based on measurement uncertainties and daily wind variances, is essential to identifying significant peaks in the LS periodogram.

135 **Reply 3.3:**

That's correct. We only performed phase analysis when the 16-day planetary wave exhibited significant amplitude.

We added confidence level information to the LS results by performing LS analysis at each altitude and extracting the 16-day wave components. This allowed us to obtain the

140 amplitude and phase as functions of height and time. A 90% confidence level was applied, and Figures 5 and 6 in the manuscript have been updated and supplemented accordingly.

[Figure]

Figure. 5 Amplitudes of 16-day waves with 90% confidence level versus height and time:
(a)(c) results of LS processing of daily-mean meridional wind perturbations; (b)(d) results of
LS processing of daily-mean zonal wind perturbations (the blank area indicates the time
period when the data coverage is less than 50%, and the black solid line represents 90%
confidence level)

[Figure]

Figure.6 16-day planetary wave phases versus height and time: (a)(c) results of LS
processing of daily-mean meridional wind perturbations; (b)(d) results of LS processing of
daily-mean zonal wind perturbations (the blank areas indicate time periods with less than
50% data coverage, and the solid black lines indicate areas with 90% confidence amplitudes).

**Comments 3.4:** Band-pass Filtering Concern:
- ○ The reviewer questions the use of a band-pass filter, which requires a Fast Fourier Transform (FFT) and thus necessitates regular gridding of the data.
- ○ Why is interpolation applied at this stage when it was carefully avoided earlier?

**Reply 3.4:**
- Band-pass filtering is a commonly used method for observing the temporal characteristics of planetary waves. It allows for a straightforward visualization of amplitude magnitude and phase trends. To facilitate comparison with previous studies that also used band-pass filtering, we applied the same method. The results from this method were used solely to observe the activity characteristics of the 16-day planetary wave; the subsequent amplitude and phase calculations were performed using harmonic fitting.

- Since the amount of missing data was minimal, we first applied interpolation to obtain complete time series before performing band-pass filtering. This has little impact on

the results. Using the FFT method, we derived the amplitude and phase and reconstructed the 16-day planetary wave time series. As shown in the figure below, the reconstruction results are compared with those from the band-pass filtering (Figure 4 in the manuscript), and they exhibit very similar amplitude distributions and phase trends.

Figure. a Results of the reconstructed time series of the 16-day planetary wave

Comments 3.5:Final Step (Harmonic Fitting & Wave Reconstruction):
- ○ Harmonic fitting could be done without prior filtering since it is already a part of the LS algorithm.
- ○ To determine the uncertainty of vertical wavelengths a Monte Carlo approach should be used in the linear fit of phase lines.
- ○ This is crucial because small errors in phase line slopes can lead to incorrect conclusions about vertical propagation direction.

**Reply 3.5:**
Adopted.
The fitting error is critical for determining the direction of vertical propagation. Based on the filtered data, we applied harmonic fitting to determine the vertical phase variation of the 16-day planetary wave. We have supplemented the analysis with fitting uncertainties and provided confidence intervals for the phase slope and vertical wavelength to demonstrate the reliability of this conclusion.
The results for the MLT and ST regions are presented in Tables 3 and 4, respectively.

Table.3 Dates, altitude ranges and calculations of significant 16-day wave in the MLT

| Date | Altitude (km) | Slope (km/rad) | Wavelength (km) |
|---|---|---|---|
| 9.3~9.18 | 90~100 | -16.37 ± 2.43 95% CI: [-21.41, -13.40] | 102.84 ± 15.25 95% CI: [84.21, 134.53] |
| 11.22~12.23 | 80~85 | 19.36 ± 10.89 95% CI: [13.34, 32.00] | 121.62 ± 59.99 95% CI: [86.52, 203.54] |
| 12.24~1.14 | 80~85 | 14.28 ± 1.08 95% CI: [13.11, 15.57] | 89.75 ± 4.31 95% CI: [82.38, 97.82] |

| Date | Altitude (km) | Slope (km/rad) | Wavelength (km) |
|---|---|---|---|
| 10.12~11.5 | 8.4~12.6 | 15.00 ± 2.28 95% CI: [12.17, 21.26]) | 94.23 ± 14.34 95%CI: [76.44, 133.59] |
| 11.7~12.2 | 6~12 | 13.50 ± 1.27 95% CI: [11.61, 16.35] | 84.80 ± 7.97 95%CI: [72.97, 102.74] |
| 12.6~12.23 | 9.6~13.2 | -9.61 ± 1.33 95% CI: [-11.35, -7.76] | 60.37 ± 8.36 95% CI: [48.78, 71.34] |
| 12.24~1.3 | 6~12 | -5.52 ± 0.42 95% CI: [-6.65, -4.91] | 34.70 ± 2.65 95% CI: [30.86, 41.76] |

Table.4 Dates, altitude ranges and calculations of significant 16-day wave in the ST

**The above provides clarification, explanation, and additional details regarding the methods and approach used in the original manuscript.**

200

For the data used in this study, planetary wave information can be obtained using the LS, FFT, or harmonic fitting methods. We applied all three approaches to calculate the amplitude and phase of the planetary waves and found that the results are largely

205     consistent across methods, as shown in the figure below.

[Figure]

**Figure. b** LS results of the zonal wind amplitude and phase

[Figure]

**Figure. c** FFT results of the zonal wind amplitude and phase

[Figure]

**Figure. d** Harmonic fitting results of the zonal wind amplitude and phase

As the manuscript involved multiple methods, we acknowledge that the use of different approaches—despite yielding consistent results—may cause confusion for readers regarding the methodology. To address this, we have standardized the analysis by using the LS method to obtain the amplitude, phase, and vertical wavelength of the 16-day planetary wave, and we have also added confidence level information.

Once again, we sincerely thank you for your valuable comments.

**References**

Day, K. A., & Mitchell, N. J. (2010). The 16-day wave in the Arctic and Antarctic mesosphere and lower thermosphere. *Atmos. Chem. Phys., 10*(3), 1461-1472. doi:https://doi.org/10.5194/acp-10-1461-2010

Huang, X. S., Huang, K. M., Zhang, S. D., Huang, C. M., Gong, Y., & Cheng, H. (2022). Extraordinary quasi-16-day wave activity from October 2013 to January 2014 with radar observations at mid-latitudes and MERRA2 reanalysis data. *Earth Planets and Space, 74*(1), 98. doi:https://doi.org/10.1186/s40623-022-01660-z

Jiang, G., Xiong, J., Wan, W., Ning, B., Liu, L., Vincent, R. A., & Reid, I. (2005). The 16-day waves in the mesosphere and lower thermosphere over Wuhan (30.6°N, 114.5°E) and Adelaide (35°S, 138°E). *Advances in Space Research, 35*(11), 2005-2010. doi:https://doi.org/10.1016/j.asr.2005.03.011

Jiang, G. Y., Xiong, J. G., Wan, W. X., Ning, B. Q., & Liu, L. B. (2005). The quasi 16-day waves in the mesosphere and lower thermosphere at Wuhan. *Chinese Journal of Space Science, 25*(1), 44-51. doi:https://doi.org/0254-6124(2005)25:1<44:WHSKMZ>2.0.TX;2-M

Lima, L. M., Batista, P. P., Clemesha, B. R., & Takahashi, H. (2006). 16-day wave observed in the meteor winds at low latitudes in the southern hemisphere. *Advances in Space Research, 38*(11), 2615-2620. doi:https://doi.org/10.1016/j.asr.2006.03.033

Lu X, & Zhang, S. D. (2005). Radiosonde observation of planetary waves in the lower atmosphere over the center China. *Chinese Journal of Space Science, 25(6)*, 529-535.

245          doi:https://doi.org/10.11728/cjss2005.06.529

Luo, Y., Manson, A. H., Meek, C. E., Meyer, C. K., Burrage, M. D., Fritts, D. C., . . . Vincent, R. A. (2002). The 16-day planetary waves: multi-MF radar observations from the arctic to equator and comparisons with the HRDI measurements and the GSWM modelling results. *Ann. Geophys., 20*(5), 691-709. doi:10.5194/angeo-20-691-2002

250    Luo, Y., Manson, A. H., Meek, C. E., Meyer, C. K., & Forbes, J. M. (2000). The quasi 16-day oscillations in the mesosphere and lower thermosphere at Saskatoon (52°N, 107°W), 1980–1996. *105*(D2), 2125-2138. doi:https://doi.org/10.1029/1999JD900979

Luo, Y., Manson, A. H., Meek, C. E., Thayaparan, T., MacDougall, J., & Hocking, W. K. (2002). The 16-day wave in the mesosphere and lower thermosphere: simultaneous observations at
255    Saskatoon (52°N, 107°W) and London (43°N, 81°W), Canada. *Journal of Atmospheric and Solar-Terrestrial Physics, 64*(8), 1287-1307. doi:https://doi.org/10.1016/S1364-6826(02)00042-1

Namboothiri, S. P., Kishore, P., & Igarashi, K. (2002). Climatological studies of the quasi 16-day oscillations in the mesosphere and lower thermosphere at Yamagawa (31.2° N, 130.6° E),
260    Japan. *Ann. Geophys., 20*(8), 1239-1246. doi:https://doi.org/10.5194/angeo-20-1239-2002

Tang, L., & Huang, C. (2016). Mid-latitude planetary waves observation from MST radar measurements in the troposphere and lower stratosphere. *Chinese Journal of Space Science, 36(2)*, 175-187. doi:https://doi.org/10.11728/cjss2016.02.175

265    Vineeth, C., Pant, T. K., Kumar, K. K., Ramkumar, G., & Sridharan, R. (2009). Signatures of low latitude–high latitude coupling in the tropical MLT region during sudden stratospheric warming. *36*(20). doi:https://doi.org/10.1029/2009GL040375

---

## Author Comment (AC2)

**Replies to the Reviewer#2**

**Dear Reviewer,**

Thank you very much for your feedback and constructive comments. I sincerely appreciate the time and effort you devoted to reviewing my manuscript. I would like to respond to the points you raised as follows (in this reply, the comments of the referee are marked in black color, and the replies in blue color):

**Comments 1:** The title does not clearly describe the article. The radar data cover the lower stratosphere and the MERRA data are not mentioned.

**Reply 1:**

**Adopted.**

We have accepted your suggestion and revised the title to: "Research on 16-day Planetary Waves in the Mid-latitude Troposphere, Stratosphere, Mesosphere, and Lower Thermosphere with Langfang Dual-frequency ST-M Radar Data and MERRA-2 Reanalysis Data."

**Comments 2:** Extracting Daily Mean Wind:

Why do the authors use the criterion of 10 hours? What are the errors of the daily wind speeds obtained with the method proposed by the authors?

**Reply 2:**

**Adopted and explanation below.**

To reduce potential biases in the estimated values caused by diurnal variations, we derived the daily mean wind fields using at least ten hourly mean wind measurements evenly distributed throughout the day and night. A preliminary estimate indicates that the standard deviation of the daily mean wind in the MLT region is within 3–6 m/s, while that in the ST region is within 0.1–2 m/s. Previous studies have also adopted similar approaches to maximize the use of observational data, such as using six hourly wind measurements to estimate the daily mean wind for analyzing the 16-day planetary wave (Luo et al., 2002), or using three hourly wind measurements for the same purpose (Jiang, Xiong, Wan, Ning, & Liu, 2005).

To improve the accuracy of the daily mean wind estimation, we raised the standard in the manuscript to include data from eighteen hourly measurements per day. As a result, the updated analysis led to five additional days of missing data; however, this does not affect the main analysis or the overall conclusions.

**Comments 3:** Detrending

The MLT winds are characterized by the strong seasonal course and strong changes in spring, sometime in autumn and during SSW (for example, in January 2024). The wind behavior model with a simple linear trend is not correct for these cases. Therefore, the 16-day wave parameters may be obtained with large errors or may be completely incorrect. I recommend removing the seasonal course first.

**Reply 3:**

**We have accepted the suggestion.**

45 We analyzed the wind field data with the seasonal course removed. While some differences in detail were observed between the results of band-pass filtering and Lomb-Scargle (LS) analysis, the overall results are improved.
These differences do not affect the main analysis or the conclusions of the study.

50 **Comments 4:** Spectral analysis
The time series of the zonal and meridional wind speeds have large gaps from month 3 to month 5 (figure 1). On the one hand, such gaps may easily distort the spectrum obtained with the Lomb-Scargle method. On the other hand, the authors fill in the gaps for their further analysis.
55 I recommend removing the data with these large gaps.
**Reply 4:**
**Adopted.**
There are data gaps in the observations from March to May, which may affect the LS spectrum. However, we decided to retain the spectral results for this period **in order to**
60 **provide a reference for the annual variation of planetary wave**. The analysis following Section 3.2 is primarily focused on the characteristics of planetary wave activity during autumn and winter.

**Comments 5:** Figure 3. Please, indicate units of the color levels. What level is significant?
65 **Reply 5:**
The color scale unit in Figure 3 is m/s, which has now been explicitly labeled in the figure. In addition, the 90% confidence level has been indicated with white dotted lines.

[Figure]

Figure 3: LS spectral analysis results of horizontal wind perturbations at different altitudes
70 over Langfang from March 2023 to February 2024: (a)(c)(e)(g) are the results of the LS spectrum analysis of the meridional wind disturbance at 96 km, 84 km, 13.8 km, and 7.2 km,

respectively; (b)(d)(f)(h) are the results of the LS spectrum analysis of the zonal wind disturbance at 96 km, 84 km, 13.8 km, and 7.2 km, respectively. (The red dashed lines from top to bottom correspond to the frequencies of 10-day and 16-day planetary waves. The white dotted line represents 90% confidence level.)

**Comments 6:** L.325-330 "No significant planetary wave activity is observed in the MLT during summer…"
How do the authors separate significant and non-significant wave activity?
**Reply 6:**
We have added confidence level information to the LS spectrum. The planetary wave amplitudes during summer are all below the 90% confidence level, indicating that the planetary wave activity in summer is not significant.
In the revised version, we have updated the height–time distribution of the 16-day wave amplitude, as shown in Figure 5.

[Figure]

Figure.5 Amplitudes of 16-day waves with 90% confidence level versus height and time: (a)(c) results of LS processing of daily-mean meridional wind perturbations; (b)(d) results of LS processing of daily-mean zonal wind perturbations (the blank area indicates the time period when the data coverage is less than 50%, and the black solid line represents 90% confidence level)

**Comments 7:** Ln. 350-365 Please, indicate errors of the vertical wavelength you found.
**Reply 7:**
We have added the errors of the planetary wave vertical wavelength in the manuscript.
We calculated the phase slope and subsequently derived the vertical wavelength. Using the bootstrap method, we estimated the errors and 95% confidence intervals for both the phase

slope and vertical wavelength. The results for the MLT and ST regions are presented in
Tables 3 and 4, respectively.

100  Table.3 Dates, altitude ranges and calculations of significant 16-day wave in the MLT

| Date | Altitude (km) | Slope (km/rad) | Wavelength (km) |
|---|---|---|---|
| 9.3~9.18 | 90~100 | -16.37 ± 2.43 95% CI: [-21.41, -13.40] | 102.84 ± 15.25 95% CI: [84.21, 134.53] |
| 11.22~12.23 | 80~85 | 19.36 ± 10.89 95% CI: [13.34, 32.00] | 121.62 ± 59.99 95% CI: [86.52, 203.54] |
| 12.24~1.14 | 80~85 | 14.28 ± 1.08 95% CI: [13.11, 15.57] | 89.75 ± 4.31 95% CI: [82.38, 97.82] |

Table.4 Dates, altitude ranges and calculations of significant 16-day wave in the ST

| Date | Altitude (km) | Slope (km/rad) | Wavelength (km) |
|---|---|---|---|
| 10.12~11.5 | 8.4~12.6 | 15.00 ± 2.28 95% CI: [12.17, 21.26]) | 94.23 ± 14.34 95%CI: [76.44, 133.59] |
| 11.7~12.2 | 6~12 | 13.50 ± 1.27 95% CI: [11.61, 16.35] | 84.80 ± 7.97 95%CI: [72.97, 102.74] |
| 12.6~12.23 | 9.6~13.2 | -9.61 ± 1.33 95% CI: [-11.35, -7.76] | 60.37 ± 8.36 95% CI: [48.78, 71.34] |
| 12.24~1.3 | 6~12 | -5.52 ± 0.42 95% CI: [-6.65, -4.91] | 34.70 ± 2.65 95% CI: [30.86, 41.76] |

**Comments 8:** Ln.367-376 It is very important for the analysis provided in this part and
105 below that the errors of the phase speeds are small enough to draw any conclusion.
**Reply 8:**
We have included the errors of the apparent vertical phase speed. The results show that the
errors do not affect the propagation direction of the apparent vertical phase speed.
Therefore, our analysis remains valid, and the conclusions of the manuscript are unchanged.
110

The errors of the apparent vertical phase speed are presented in Tables 5 and 6.

Table.5 Relationship between m, k, apparent phase speed $c_{pz,a}$ and $c_{gz}$ of the significant 16-day wave in the ST

| Date | Altitude (km) | Positive and negative of $m$ | $k$ | $c_{pz,a}$ (cm/s) | Direction of $c_{pz,a}$ | Direction of $c_{gz}$ |
|---|---|---|---|---|---|---|
| 10.12~11.5 | 8.4~12.6 | + | 3 | 6.82 ± 1.04 95%CI: [5.53, 9.66] | ↑ | ↑ |
| 11.7~12.2 | 6~12 | + | 3 | 6.13 ± 0.58 95%CI: [5.28, 7.43] | ↑ | ↑ |
| 12.6~12.23 | 9.6~13.2 | - | -2 | -4.37 ± 0.60 95%CI: [-5.16, -3.53] | ↓ | ↑ |
| 12.24~1.3 | 6~12 | - | -2 | -2.51 ± 0.19 95%CI: [-3.02, -2.23] | ↓ | ↑ |

115  Table.6 Relationship between m, k, apparent phase speed $c_{pz,a}$ and $c_{gz}$ of the significant 16-day wave in the MLT

| Date | Altitude (km) | Positive and negative of $m$ | $k$ | $c_{pz,a}$ (cm/s) | Direction of $c_{pz,a}$ | Direction of $c_{gz}$ |
|---|---|---|---|---|---|---|
| 9.3~9.18 | 90~100 | - | -3 | -7.44±1.10 95%CI: [-9.73,-6.09] | ↓ | ↑ |
| 11.22~12.23 | 80~85 | + | 3 | 8.80±4.34 95%CI: [6.26,14.72] | ↑ | ↑ |
| 12.24~1.14 | 80~85 | + | 3 | 6.50±0.31 95%CI: [6.25,6.12] | ↑ | ↑ |

**Comments 9:** Ln.380  $Q_y < 0$ is not sufficient for the instability.
**Reply 9:**
Accepted.
We have removed this sentence.

**Comments 10:** Ln.380-383 The authors use the result of the quasi-geostrophic theory.
The conclusion from eq.3 is not correct. The vertical phase speed is opposite to the vertical
group speed in the coordinate system that moves with the zonal flow (Qy > 0). If one takes
into account the background zonal flow $U_0$, then the result will be complex and will depend
on $U_0$.
By the way, the authors should explain the notation in the equation and provide a reference.
**Reply 10:**
Adopted.
We have added the $1/4H^2$ term in eq.3. There was a mistake in the value of $c_p$ given
in the manuscript. We have added descriptions and explanations of the apparent
vertical phase speed $c_{pz,a} = \sigma/m$ and the intrinsic vertical phase speed $c_{pz,i} = \omega/m$.

Our results show that the apparent vertical phase speed undergoes a reversal from
upward to downward propagation, whereas the vertical group velocity consistently
propagates upward without any change in direction.

Although the intrinsic vertical phase speed and the vertical group velocity propagate in
opposite directions, this relationship does not hold for the apparent vertical phase
speed. The apparent vertical phase speed is influenced in a more complex way by the
background wind.

The descriptions and explanations of the apparent vertical phase speed $c_{pz,a} = \sigma/m$ and the
intrinsic vertical phase speed $c_{pz,i} = \omega/m$ are as follows:
The perturbation potential vorticity equation for wave motion on a β plane(Salby, 1995):

$$(\frac{\partial}{\partial t} + U\frac{\partial}{\partial x})[\frac{\partial^2}{\partial x^2} + \frac{\partial^2}{\partial y^2} + \frac{\partial}{\partial z}(\frac{f_0^2}{N^2}\frac{\partial}{\partial z})]\psi' + \beta\frac{\partial\psi'}{\partial x} = 0 \tag{1}$$

where U represents the background zonal wind, z refers to log-pressure height, $f_0$ is the
Coriolis parameter, $N$ is the Brunt–Väisälä frequency, $\psi'$ is the geostrophic streamfunction,
and $\beta$ is the planetary vorticity gradient.
Since coefficients are constant, we consider solutions of the form $e^{[(z/2H)+i(kx+ly+mz-\sigma t)]}$,
where $H$ is the atmospheric density scale height, $k$、$l$、$m$ are the zonal, meridional, and
vertical wavenumbers, respectively, and $\sigma$ is the apparent frequency. Substituting into the
above equation gives the **apparent frequency $\sigma$**:

$$\sigma = Uk - \frac{\beta k}{k^2+l^2+(\frac{f_0^2}{N^2})(m^2+\frac{1}{4H^2})} \tag{2}$$

The **intrinsic frequency $\omega$** is given by:

$$\omega = \sigma - Uk = -\frac{\beta k}{k^2+l^2+(\frac{f_0^2}{N^2})(m^2+\frac{1}{4H^2})} \tag{3}$$

The data measured by the ST-M radar, after frequency-domain processing, yield the apparent frequency. From this, the **apparent vertical phase speed** can be further derived as:

$$c_{pz,a} = \frac{\sigma}{m} \tag{5}$$

The **intrinsic vertical phase speed** is given by:

$$c_{pz,i} = \frac{\omega}{m} \tag{6}$$

For the 16-day wave, $\sigma$为$\frac{2\pi}{16} rad/day$, and the direction of the apparent phase speed depends on the sign of $m$.

The **vertical group velocity** is given by:

$$c_{gz} = \frac{\partial \omega}{\partial m} = \frac{2\beta m \, k(\frac{f_0^2}{N^2})}{[k^2 + l^2 + (\frac{f_0^2}{N^2})(m^2 + \frac{1}{4H^2})]^2} = (\frac{f_0^2}{N^2})\frac{2m\omega^2}{\beta k} \tag{7}$$

The direction of the group velocity depends on the sign of $mk$.

**Comments 11:** L.385 "Fig.5. Assuming that the frequency-wavenumber spectrum in the MLT is consistent with that at the 79 km altitude, the dominant wavenumber for each time was selected as the wavenumber for that period. "
Why are the spectra consistent? The MERRA-2 data are given at the model level, but the MR winds are given at altitudes. The difference between the true heights may be significant.

**Reply 11:**
The wavenumber can vary with altitude. In the ST region, we obtained the frequency–wavenumber spectra at different altitudes using MERRA-2 reanalysis data. However, due to the lack of other data sources, we were unable to obtain the frequency–wavenumber spectra in the MLT region. As a simple reference, we used the spectrum at approximately 79 km derived from MERRA-2 reanalysis data.
This approximation may be inaccurate for regions with large altitude differences and is only intended as a possible explanation for the observed reversal in phase propagation direction in the MLT region.

**Comments 12:** There is no confidence that oscillations presented in Fig.5 are statistically significant.

**Reply 12:**
We used the **Monte Carlo method** to estimate the confidence levels of the wavenumber. The regions outlined by solid white lines in the figure indicate wavenumbers with a 90% confidence level.

[Figure]

Figure.7 Frequency-wavenumber spectrum with a 90% confidence level: (a), (b), and (c) are the frequency-wavenumber spectra from 1.2 to 75.6 km for the periods September 3 to October 4, October 12 to December 2, and December 6 to January 14, respectively (negative wavenumber indicates westward wave propagation).

**Comments 13:** The aim of this part is "To investigate the location of the wave source in the ST" and "the relationship between the planetary waves in the two regions".
The analysis is confined in latitude and longitude to the region where the radar is located. Therefore, the authors (and the readers) do not know how 16-day waves propagate in the neighboring region. Hence, the authors can't really reach their aim.
Additional note, the real atmospheric 16-day waves are transient, their amplitudes are changing with time as observed. The theory used in this part does not work for such waves. I propose to find and plot the E-P flux for the waves.
**Reply 13:**
**We have made corresponding adjustments to the objective of this study.**
The aim of the paper is to use ST-M radar observations to reveal the characteristics of 16-day planetary wave activity in the ST and MLT regions above Langfang, and to attempt to explain a newly observed phenomenon—namely, the reversal of the vertical phase propagation direction of the 16-day wave in the ST region.

The ST-M radar at the Langfang observation station is a newly developed instrument for detecting wind fields in the ST and MLT regions over mid-latitudes. Observational data reveal that the 16-day wave simultaneously exists in both regions above Langfang.

To explore whether there is a connection between the planetary waves observed in the ST

and MLT regions above Langfang, we introduced MERRA-2 reanalysis data to extract the zonal wavenumber and analyze the background wind conditions associated with the upward propagation of the 16-day wave. This provides a useful reference for understanding the relationship between planetary waves in the ST and MLT regions over Langfang.

The identification of planetary wave sources and the study of their propagation require global data and the application of Eliassen–Palm (E-P) flux analysis. In future work, we plan to conduct a dedicated analysis of planetary wave propagation across different longitudes and atmospheric layers by utilizing reanalysis datasets such as MERRA-2 and employing methods including E-P flux analysis.

**Comments 14:** Ln.465 "The quasi-16-day and quasi-10-day waves dominate in both the ST and MLT regions" – this conclusion may be a result of the 32-day segment used for the analysis and a linear trend model. The waves with shorter periods are just averaged over the segment and their transient behavior is not taken into account.

**Reply 14:**
We have revised it to: "The quasi-16-day and quasi-10-day waves behavior obviously in both the ST and MLT regions."

**Comments 15:** Conclusions (2) and (3) repeat the first one. The errors of phase speeds are not clear. Therefore, the statements about their changes are not supported in the text.

**Reply 15:**
We have added the estimation uncertainty of the apparent vertical phase speed, which provides additional support for our analysis. The supplementary results are presented in **Comments 8**.

**Comments 16:** Ln.475-479 Please, see above. The authors' statements are incorrect.

**Reply 16:**
After correcting the mistake in the value of $c_p$, our conclusion remains valid. The explanation is as follows:

In **Comments 10**, we discussed the relationships among the zonal wavenumber k, vertical wavenumber m, apparent vertical phase speed $c_{pz,a}$, intrinsic vertical phase speed $c_{pz,i}$ and vertical group velocity $c_{gz}$. The results are as follows:

The direction of the apparent vertical phase speed $c_{pz,a}$ depends on the sign of the vertical wavenumber $m$, while the direction of the vertical group velocity $c_{gz}$ depends on the sign of $mk$.

Although the intrinsic vertical phase speed $c_{pz,i}$ and the vertical group velocity $c_{gz}$ propagate in opposite directions, this relationship does not apply to the apparent vertical phase speed $c_{pz,a}$, which is influenced in a more complex manner by the background wind.

**Comments 17:** Ln. 480 Conclusion (4). This conclusion does not have a solid support from the analysis as it is noted above. The sign of speeds, the wavenumber estimations are in question.

**Reply 17:**

**After distinguishing between the apparent phase speed and the intrinsic phase speed, we maintain that the conclusion is valid. The reasons are as follows:**

(1) As discussed in Comments 10, the intrinsic vertical phase speed $c_{pz,i}$ and the vertical group velocity $c_{gz}$ propagate in opposite directions. However, this relationship does not apply to the apparent vertical phase speed $c_{pz,a}$, which is influenced in a more complex way by the background wind.

(2) In Comments 8, we supplemented the analysis with errors of the apparent vertical phase speed. The results show that the errors do not affect the propagation direction of the apparent vertical phase speed.

(3) In Comments 12, we used the Monte Carlo method to estimate the 90% confidence level of the zonal wavenumber $k$.

Based on the above conclusions and results, and in combination with the zonal background wind shown in Figure 6 of the manuscript, we analyzed the apparent vertical phase speed $c_{pz,a}$, the intrinsic vertical phase speed $c_{pz,i}$ and the vertical group velocity $c_{gz}$ of the significant 16-day waves in the ST region during November 7–December 2 and December 6–23.

The apparent frequency $\sigma$ of the significant 16-day wave, $\sigma \approx 4.55 * 10^{-6} \, rad/s$.

The intrinsic frequency $\omega$, $\omega = \sigma - Uk$.

During November 7 to December 2, the zonal background wind was approximately $U \approx 38 m/s$, with a zonal wavenumber of 3, $k \approx 6.10 * 10^{-7} \, rad/m$, and $\omega \approx -2.23 * 10^{-5} \, rad/s$. In this case, since ω<0 and m>0, the intrinsic vertical phase speed $c_{pz,i} < 0$, indicating that the energy of the 16-day wave propagates upward.

During December 6 to December 23, the zonal background wind was approximately $U \approx 44 m/s$, with a zonal wavenumber of -2, $k \approx -4.07 * 10^{-7} \, rad/m$, and $\omega \approx 2.24 * 10^{-5} \, rad/s$. In this case, since ω>0 and m<0, the intrinsic vertical phase speed $c_{pz,i} < 0$, indicating that the energy of the 16-day wave propagates upward.

**These results further confirm that the apparent vertical phase speed $c_{pz,a}$ of the significant 16-day wave underwent a reversal. However, due to the change in the zonal wavenumber k, the direction of energy propagation remained unchanged. Before and after the reversal of $c_{pz,a}$, the energy of the 16-day wave consistently propagated upward.**

**Comments 18:** Please, directly indicate height intervals and/or time intervals on each plot.

**Reply 18:**

We have indicated the height and/or time intervals in the figure.

To improve clarity for readers, we simplified the data processing approach by using the Lomb-Scargle method to derive the amplitude, phase, and vertical wavelength of the 16-day planetary wave, and we also added information on the significance level.

310

Once again, we sincerely thank you for your valuable comments.

**References**

315 Jiang, G. Y., Xiong, J. G., Wan, W. X., Ning, B. Q., & Liu, L. B. (2005). The quasi 16-day waves in the mesosphere and lower thermosphere at Wuhan. *Chinese Journal of Space Science, 25*(1), 44-51. doi:https://doi.org/0254-6124(2005)25:1<44:WHSKMZ>2.0.TX;2-M

Luo, Y., Manson, A. H., Meek, C. E., Meyer, C. K., Burrage, M. D., Fritts, D. C., . . . Vincent, R. A. (2002). The 16-day planetary waves: multi-MF radar observations from the arctic to 320 equator and comparisons with the HRDI measurements and the GSWM modelling results. *Ann. Geophys., 20*(5), 691-709. doi:https://doi.org/10.5194/angeo-20-691-2002

Salby, M. L. (1995). *Fundamentals of atmospheric physics*: International Geophysics.